# A directional surface reflectance climatology determined from TROPOMI observations

Lieuwe G. Tilstra[1], Martin de Graaf[1], Victor J. H. Trees[1], Pavel Litvinov[2], Oleg Dubovik[3], and Piet Stammes[1]

[1]Royal Netherlands Meteorological Institute (KNMI), De Bilt, the Netherlands
[2]GRASP SAS, Remote Sensing Developments, 59260 Lezennes, France
[3]Laboratoire d'Optique Atmosphérique, CNRS/Université de Lille, Villeneuve-d'Ascq, 59650, France

**Correspondence:** Lieuwe G. Tilstra (tilstra@knmi.nl)

**Abstract.** In this paper we introduce a spectral surface reflectivity climatology based on observations made by the TROPOMI instrument on board the Sentinel-5P satellite. The database contains the directionally dependent Lambertian-equivalent reflectivity (DLER) of the Earth's surface for 21 wavelength bands ranging from 328 nm to 2314 nm and for each calendar month. The spatial resolution of the database grid is $0.125° \times 0.125°$. A recently developed cloud shadow detection technique is implemented to avoid dark scenes due to cloud shadow. In the database, the anisotropy of the surface reflection is described using a third-order parameterisation of the viewing angle dependence. The viewing angle dependence of the DLER is analysed globally and for a selection of surface type regions. The dependence is found to agree with the viewing angle dependence found in the GOME-2 surface DLER database. Differences exist, related to the actual solar position. On average, the viewing angle dependence in TROPOMI DLER is weaker than for GOME-2 DLER, but still important.

Validation of the new database was first performed by comparison of the non-directional TROPOMI surface LER with heritage LER databases based on GOME-1, OMI, SCIAMACHY, and GOME-2 data. Agreement was found within 0.002–0.02 in the UV-VIS (below 500 nm), up to 0.003 in the NIR (670–772 nm), and below 0.001 in the SWIR (2314 nm). These performance numbers are dominated by the performance over ocean, but they are in most cases also representative for land surfaces. For the validation of the directional TROPOMI surface DLER we made use of comparison with MODIS surface BRDF for a selection of surface type regions. In all cases the DLER performed significantly better than the traditional LER and we found good agreement with MODIS surface BRDF.

The TROPOMI surface DLER database is a clear improvement on previous surface albedo databases and can be used as input not only for satellite retrievals from TROPOMI observations, but also for retrievals from observations from other polar-orbiting satellite instruments provided that their equator crossing time is close to that of TROPOMI. The algorithm that is introduced in this paper can be used for the retrieval of surface reflectivity climatologies from other polar satellite missions as well, including OLCI on the Sentinel-3 satellites, Sentinel-5 and 3MI on the MetOp-SG-A1 satellite to be launched in 2025, and the future CO2M mission.

# 1 Introduction

The reflectivity of the Earth's surface is important input for many satellite retrievals of atmospheric composition. Examples of
retrievals for which this is the case are the retrievals of ozone ($O_3$), nitrogen dioxide ($NO_2$), methane ($CH_4$), formaldehyde
($CH_2O$), bromine oxide (BrO), water vapour ($H_2O$), carbon monoxide (CO) and carbon dioxide ($CO_2$), and of cloud and
aerosol information. In many retrieval codes, the surface is described as a Lambertian reflector, meaning that the surface
reflection is assumed to be isotropic. This is a simplified approach which may be justified in some cases, but certainly not in
all. For instance, Lorente et al. (2018) show that relying on traditional Lambertian surface reflection databases can result in
errors of a factor of two in the assumed surface reflection for vegetated surfaces – at least for certain viewing geometries – due
to their lack of directional dependence.

Generally speaking, it is better to describe the surface reflection as a function of incidence and reflection angles, using a
bi-directional reflectance distribution function (BRDF) (Nicodemus et al., 1992; Schaepman-Strub et al., 2006). The MODIS
surface BRDF database (Gao et al., 2005), an established product, is available for land surfaces and is widely used. However,
inserting a BRDF where the radiative transfer code expects a Lambertian surface reflection would lead to errors, especially for
the shorter wavelengths (Tilstra et al., 2021, Sect. 3.3). Apart from that, the MODIS BRDF wavelength bands are not positioned
spectrally such that they can support all atmospheric retrievals mentioned above.

Recently, a number of new databases have appeared that provide Lambertian surface albedos which include a directional de-
pendence of the albedo values. The geometry-dependent surface Lambertian-equivalent reflectivity (GLER) (Qin et al., 2019)
uses MODIS surface BRDF information and converts it to a Lambertian surface albedo at 466 nm for satellite footprints of
the OMI instrument, respecting the viewing and solar directions of the OMI observations. This is done for the land-covered
footprints. For the water-covered footprints model calculations are used (Fasnacht et al., 2019). The geometry-dependent ef-
fective Lambertian-equivalent reflectivity (GE_LER) database (Loyola et al., 2020) also provides Lambertian surface albedos
with a directional dependence. The surface albedo in this case is derived from level-1 data from GOME-2 or TROPOMI. The
GE_LER provides daily maps of the surface properties for land, ocean, and snow/ice in one database. The directionally depen-
dent Lambertian-equivalent reflectivity (DLER) derived from GOME-2 (Tilstra et al., 2021) is a monthly surface reflectivity
climatology derived from GOME-2 level-1 observations. This database also provides the directional dependence of the surface
reflectivity but via a parameterisation of the viewing angle dependence. That is, the provided directional surface albedo is not
only available for a specific satellite footprint or for a specific satellite instrument, but for any observation from any satellite
instrument provided that the overpass time of the satellite instrument is close to that of the GOME-2 orbit (09:30 LT).

Because the surface albedos provided by the GLER, GE_LER, and DLER databases are Lambertian (despite their directional
dependence), those albedos can be used as input for radiative transfer codes that rely on Lambertian surface reflection. The
addition of directionality is in all cases a large improvement on traditional non-directional surface albedo databases based
on, for instance, TOMS (Herman and Celarier, 1997), GOME (Koelemeijer et al., 2003), OMI (Kleipool et al., 2008), and
SCIAMACHY (Tilstra et al., 2017). However, a specific TROPOMI DLER database exploiting the small footprint size of

TROPOMI did not exist, and the existing GOME-2 DLER database cannot be used as input for TROPOMI retrieval algorithms because of the different overpass time of the GOME-2 orbit.

This paper introduces a new surface albedo database based on measurements performed by the Tropospheric Monitoring Instrument (TROPOMI) (Veefkind et al., 2012). The database contains the TROPOMI surface DLER retrieved for 21 wavelength bands ranging from 328 to 2314 nm on a two-dimensional grid of the Earth's surface with a spatial resolution of $0.125° \times 0.125°$. This spatial resolution is higher than that of the heritage databases. The database includes a directional description of the surface reflection, as does the GOME-2 surface DLER database, but the angular dependence is improved with a third-order instead of a second-order parameterisation. The use of the TROPOMI instrument allows a wavelength band in the SWIR (at 2314 nm). The retrieval approach that was followed relies on heritage from previous climatologies, but has been improved in a number of ways. The retrieval algorithm employs a combination of active and statistical cloud filtering which results in less cloud contamination in the database. Also, cloud shadows are not ignored but removed using a recently developed cloud shadow detection technique (Trees et al., 2022). For the generation of the DLER database we used TROPOMI level-1b data version 2.1.0 (doi:10.5270/S5P-kb39wni). For the validation study we used accuracy requirements on the DLER of 0.03+10% (0.03 plus 10% of the value, below 500 nm) and 0.02+10% (above 670 nm). These target requirements were taken from the final report of ESA's Sentinel-5p+ Innovation AOD/BRDF project (Litvinov et al., 2022).

This paper has the following structure. Section 2 briefly introduces the TROPOMI instrument. Section 3 introduces the theory behind (directional) Lambertian surface reflection. In Sect. 4 the algorithm set-up and the retrieval code are discussed. Examples of the anisotropic surface reflectivity observed by TROPOMI, and of cloud and aerosol features in the database, are presented in Sect. 5. Section 6 presents the results of a validation of the new TROPOMI surface DLER database. The paper ends in Sect. 7 with conclusions and an outlook to the future.

## 2  Description of TROPOMI

The TROPOMI instrument (Veefkind et al., 2012) was launched on 13 October 2017 on board the Sentinel-5 Precursor (S5P) satellite. TROPOMI is the only instrument on S5P. The S5P satellite was brought into a near-polar, Sun-synchronous orbit, on average 824 km above the Earth's surface, with an orbital period of about 101 minutes. The local equator crossing time of the S5P satellite is 13:30 LT for the ascending node, which is very close to that of the Aura satellite hosting TROPOMI's predecessor OMI (Levelt et al., 2006).

TROPOMI is a nadir-looking spectrometer equipped with two-dimensional CCD and CMOS detectors (Ludewig et al., 2020). The spectral domain is observed in one dimension and an across-track slice of the Earth in the other. The orbit is scanned in by the forward movement of the satellite while the instrument is carrying out its measurements every 1.08 s. The footprint size was $7.2 \times 3.6 \text{ km}^2$ from the start of the mission until 6 August 2018, on which day the footprint size was reduced to $5.6 \times 3.6 \text{ km}^2$. The orbit swath is 2600 km wide, which allows global coverage in one day.

As a spectrometer, TROPOMI covers the ultraviolet-visible wavelength range (UV-VIS, 267–499 nm), the near-infrared wavelength range (NIR, 661–786 nm), and the shortwave infrared wavelength range (SWIR, 2300–2389 nm). This is a large

improvement compared to OMI, which only observes the UV-VIS wavelength range (270–500 nm). The extended wavelength range of TROPOMI (compared to OMI) makes it possible to retrieve additional trace gases, for example CO and $CH_4$, and to perform retrievals of cloud and aerosol properties using the $O_2$-A and $O_2$-B absorption bands.

The radiometric calibration of the TROPOMI instrument has been improved a number of times since its launch. The latest version of the level-1b data (v2.1.0) includes, amongst other things, a correction for instrument degradation. An issue in the radiometric calibration of spectral bands 3 and 4 (Tilstra et al., 2020) has been resolved in this version. More information about the TROPOMI instrument, its calibration, and the products derived from it can be found in Kleipool et al. (2018); Ludewig et al. (2020) and in Veefkind et al. (2012).

## 3 Theory

### 3.1 Definitions

The top of the atmosphere (TOA) reflectance $R$ is defined in this paper as follows:

$$R(\mu, \mu_0, \phi, \phi_0) = \frac{\pi I(\mu, \mu_0, \phi, \phi_0)}{\mu_0 E_0}. \tag{1}$$

In Eq. (1), $I$ is the Earth radiance at the TOA, given in units of $Wm^{-2}sr^{-1}nm^{-1}$. The symbol $E_0$ refers to the extraterrestrial solar irradiance perpendicular to the beam, given in units of $Wm^{-2}nm^{-1}$. We have $\mu_0 = \cos\theta_0$, where $\theta_0$ represents the solar zenith angle. For the viewing direction we have in a similar way $\mu = \cos\theta$, with $\theta$ the viewing zenith angle. The viewing and solar azimuth angles are denoted by $\phi$ and $\phi_0$, respectively.

### 3.2 Lambertian-equivalent reflectivity

For clear-sky situations, the following relationship is known to be valid (Chandrasekhar, 1960):

$$R(\mu, \mu_0, \phi - \phi_0, A_s) = R^0(\mu, \mu_0, \phi - \phi_0) + \frac{A_s T(\mu, \mu_0)}{1 - A_s s^\star}. \tag{2}$$

The quantity $R^0$ is the path reflectance, which represents the contribution of the atmosphere to the TOA reflectance in the absence of surface reflection. That is, it is the reflectance of a Rayleigh atmosphere which is bounded below by a black surface. The second term in Eq. (2) represents the contribution of the surface to the TOA reflectance. It contains the (Lambertian) surface albedo $A_s$, the total transmission of the atmosphere $T$, and the spherical albedo $s^\star$ of the atmosphere illuminated from below by light reflected by the surface.

From a measured TOA reflectance $R^{obs}$, the (Lambertian) surface albedo $A_s$ can then be determined using Eq. (2):

$$A_s = \frac{R_\lambda^{obs} - R_\lambda^0}{T_\lambda(\mu, \mu_0) + s_\lambda^\star(R_\lambda^{obs} - R_\lambda^0)}. \tag{3}$$

The parameter $A_s$ found in this approach is the so-called Lambertian-equivalent reflectivity (LER) of the surface.

## 3.3 Directional surface LER

The Lambertian-equivalent reflectivity (LER) of the surface as defined in Sect. 3.2 is in principle meant to be an isotropic property. In most situations in reality surface reflection is not isotropic. Indeed, for a better description of surface reflection one needs to use a BRDF, which is able to fully describe the dependence on the angles associated with the radiation reaching and leaving the surface. However, BRDFs cannot be used in algorithms that assume a Lambertian surface.

In a recent paper (Tilstra et al., 2021), the concept of directionally dependent Lambertian-equivalent reflectivity (DLER) was introduced. The surface DLER can be defined as a Lambertian surface albedo (LER) retrieved as a function of the viewing angle $\theta_v$, which in this paper is defined as:

$$
\theta_v = \begin{cases} -\theta & , & \text{for the east viewing direction} \\ \theta & , & \text{for the west viewing direction} \end{cases}
\tag{4}
$$

The dependence on $\theta_0$ and $\phi - \phi_0$ is effectively linked to the combination of $\theta_v$ and the geographical latitude via the orbit of the satellite instrument. This is a good approximation over the course of a single month. Because the DLER, like the traditional non-directional LER, is a Lambertian property, it can still be used as input in radiative transfer calculations in which Lambertian surface reflection is applied. The strength of the DLER is that it describes the anisotropy of the surface reflection in a very concise manner.

## 4 Algorithm set-up

The algorithm set-up is close to algorithm set-ups described earlier (Tilstra et al., 2017, 2021): The reflectance spectrum of each satellite footprint is transformed into a set of reflectances for 21 carefully defined wavelength bands (see Sect. 4.1). Next, following Sect. 4.2, the band reflectances are converted into scene LER values by applying the atmospheric correction described in Sect. 3.2 and expressed by Eq. (3). After that, all scene LER observations that belong to a certain calendar month (e.g. March) are distributed onto a latitude/longitude grid. The surface LER is then retrieved for each grid cell from the distribution of the scene LER values, in the way described in Sect. 4.6. During the gridding of the data, active and statistical cloud filtering (see Sect. 4.3) and filtering for absorbing aerosol (see Sect. 4.5) is applied. Cloud shadows are also filtered out (see Sect. 4.4) as well as data affected by solar eclipses. The additional steps needed to retrieve DLER, the directional LER, are discussed in Sect. 4.7. Finally, various post-processing corrections handle issues like cloud contamination and gaps due to polar night (see Sect. 4.8).

There are quite some differences of the algorithm compared to the previous algorithms. One important difference is the use of active cloud screening instead of relying purely on statistical cloud screening (see Sect. 4.3). This improves the quality of the database, especially for regions where cloud contamination is an issue. Note that the algorithm still uses statistical cloud screening as a second-stage cloud filter. As a result of the different cloud filtering procedure, the algorithm does not provide the so-called MIN-LER and MODE-LER fields that were part of earlier databases (e.g. Kleipool et al., 2008; Tilstra et al., 2017). Instead, the algorithm distinguishes between two types of grids representing snow/ice and snow/ice-free conditions (see

Sect. 4.6). Another improvement is the cloud shadow detection and filtering (see Sect. 4.4). More detailed information about the algorithm set-up can be found in the Algorithm Theoretical Baseline Document (ATBD) (Tilstra et al., 2023).

## 4.1 Wavelength bands

First, we decide on the best set of wavelength bands for the DLER database. In Table 1, the central wavelength and the bandwidth of the wavelength bands are presented, along with the instrument channel, or TROPOMI band, from which the wavelength bands originate. The 21 wavelength bands were mostly selected based on heritage considerations, i.e. their selection was motivated by their presence in the heritage databases mentioned in the introduction of this paper. For the calculation of the reflectance bands from the reflectance spectrum a triangular weighting function $w$ is used, which is defined in the following way:

$$w_i^j = \begin{cases} 1 - \dfrac{|\lambda_i - \lambda_j^{\mathrm{c}}|}{\omega_j} & , \quad \text{for } |\lambda_i - \lambda_j^{\mathrm{c}}| \leq \omega_j \\ 0 & , \quad \text{for } |\lambda_i - \lambda_j^{\mathrm{c}}| > \omega_j \end{cases} \tag{5}$$

In Eq. (5), $\lambda_i$ denotes the wavelength associated to detector pixel $i$, the parameter $\lambda_j^{\mathrm{c}}$ refers to the central wavelength of wavelength band $j$, as indicated in the third row of Table 1, and the parameter $2\omega_j$ is the full bandwidth of wavelength band $j$, as indicated in the fourth row of Table 1.

## 4.2 Calculating the scene LER

From the TOA reflectances, we compute the scene LER using Eq. (3). The parameters $R^0$, $T$ and $s^\star$ needed for that are calculated by the radiative transfer model "Doubling-Adding KNMI" (DAK) (de Haan, 1987; Stammes, 2001) and stored in look-up tables (LUTs) in a manner described in Tilstra et al. (2021). For the majority of the wavelength bands, monochromatic calculations of the reflectances suffice. This is indicated in the fifth row of Table 1 with an "M". Monochromatic calculations are in these cases justified because the wavelength bands were positioned in the continuum parts of the spectrum. For a number of the wavelength bands, however, spectral calculations are needed. This is indicated in the fifth row of Table 1 with an "S". These wavelength bands are affected, to a small degree, by absorption of near-by absorption bands. Table 1 also lists the atmospheric species which are taken into account by the radiative transfer calculations. For instance, the wavelength bands at 697 and 712 nm, which are relevant to retrievals using the $O_2$-B band (see e.g. Desmons et al., 2019), are under the influence of absorption by oxygen and water vapour.

## 4.3 Cloud screening

Cloud screening on the scene LER is performed in two ways. First, active cloud filtering is applied by using cloud information from the S5P NPP-VIIRS cloud information product, which is derived from observations by the Visible Infrared Imaging Radiometer Suite (VIIRS) instrument located on the Suomi National Polar-orbiting Partnership (Suomi NPP) satellite. The Suomi NPP satellite is in an orbit close to that of the S5P satellite, with a relatively small overpass time difference of 3

**Table 1.** Definition of the wavelength bands and details of the radiative transfer calculations for atmospheric correction.

| Wavelength band | 328 | 335 | 340 | 354 | 367 | 380 | 388 | 402 | 416 | 425 | 440 |
|---|---|---|---|---|---|---|---|---|---|---|---|
| Instrument channel | 3 | 3 | 3 | 3 | 3 | 3 | 3 | 4 | 4 | 4 | 4 |
| Central wavelength (nm) | 328.0 | 335.0 | 340.0 | 354.0 | 367.0 | 380.0 | 388.0 | 402.0 | 416.0 | 425.0 | 440.0 |
| Bandwidth (nm) | 1.0 | 1.0 | 1.0 | 1.0 | 1.0 | 1.0 | 1.0 | 1.0 | 1.0 | 1.0 | 1.0 |
| Spectral/monochromatic | S | M | M | M | M | M | M | M | M | M | M |
| $O_3$ absorption | + | + | + | + | + | + | + | + | + | + | + |
| $NO_2$ absorption | + | + | + | + | + | + | + | + | + | + | + |
| $O_2$-$O_2$ absorption | + | + | + | + | + | + | + | + | + | + | + |
| $O_2$ absorption | – | – | – | – | – | – | – | – | – | – | – |
| $H_2O$ absorption | – | – | – | – | – | – | – | – | – | – | – |

| Wavelength band | 463 | 494 | 670 | 685 | 697 | 712 | 747 | 758 | 772 | 2314 |
|---|---|---|---|---|---|---|---|---|---|---|
| Instrument channel | 4 | 4 | 5 | 5 | 5 | 5 | 6 | 6 | 6 | 7 |
| Central wavelength (nm) | 463.0 | 494.0 | 670.0 | 685.0 | 696.97 | 712.7 | 747.0 | 758.0 | 772.0 | 2314.0 |
| Bandwidth (nm) | 1.0 | 1.0 | 1.0 | 1.0 | 0.3 | 0.3 | 1.0 | 1.0 | 1.0 | 0.5 |
| Spectral/monochromatic | M | M | M | M | S | S | M | S | S | M |
| $O_3$ absorption | + | + | + | + | + | + | + | + | + | + |
| $NO_2$ absorption | + | + | + | + | + | + | + | + | + | + |
| $O_2$-$O_2$ absorption | + | + | + | + | + | + | + | + | + | + |
| $O_2$ absorption | – | – | – | – | + | + | – | + | + | – |
| $H_2O$ absorption | – | – | – | – | + | + | – | – | – | – |

The reflectance calculations are performed using spectral band integration or monochromatically. For all wavelength bands absorption by ozone, $NO_2$, and $O_2$-$O_2$ is included. Absorption by oxygen is included for the 758 and 772-nm wavelength bands. For the 697 and 712-nm wavelength bands absorption by oxygen and water vapour is included.

minutes. The S5P NPP-VIIRS product can provide accurate cloud information for each of the TROPOMI footprints. We use the number of VIIRS observations which were confidently clear ($N_{c.clr}$), probably clear ($N_{p.clr}$), probably cloudy ($N_{p.cld}$), and confidently cloudy ($N_{c.cld}$) (Siddans, 2016) to calculate a geometrical cloud fraction $c_f$:

$$c_f = \frac{N_{c.cld}}{N_{c.clr} + N_{p.clr} + N_{c.cld} + N_{p.cld}}. \tag{6}$$

This definition is different from previous definitions used in e.g. Tilstra et al. (2020). Because of this, the cloud filtering is less strict and only the undisputed cases of cloud cover are removed. The reason for selecting a less strict filtering is that too many scene LER observations were deleted as a result of incorrect cloud flagging because of, for instance, the 3-minute time difference between Suomi NPP and S5P. Also, this first filtering is primarily used to reduce the amount of data involved and to remove the most obvious cases of cloud cover. The threshold for $c_f$ was set to 0.03. After this initial active cloud filtering,

statistical cloud filtering is applied, in the manner explained in Sect. 4.6.

## 4.4 Cloud shadow screening

Cloud shadows can significantly reduce the reflectance measured by TROPOMI and can lower the retrieved scene LER such that it can even become negative (Trees et al., 2022, Fig. 5). Cloud shadow was not a serious problem for earlier surface albedo databases based on, for instance, SCIAMACHY or GOME-2, because of the large footprint sizes of the measurements. The fraction of the footprint area covered in shadow is therefore relatively low for these instruments. To be able to filter out scene LER observations affected by cloud shadows we implemented the cloud shadow detection algorithm DARCLOS by Trees et al. (2022). This algorithm is based on a two-step approach. In the first step, a potential cloud shadow flag (PCSF) is calculated. This flag is based on the geometrical situation at hand: the position of the Sun, the viewing direction, and the height of the cloud responsible for the cloud shadow. The PCSF is designed to filter out cloud shadows, but it does it very rigorously, thereby throwing away too many observations. For that reason, we do not use it as a direct filter, but use it in combination with the second step of the cloud shadow filtering approach.

In this second step, the spectral cloud shadow flag (SCSF) is calculated. The SCSF is determined using a contrast parameter which is based on the measured scene LER value and the expected surface DLER value:

$$\Gamma(\lambda) = \frac{A_{\text{scene}}(\lambda) - A_{\text{DLER}}(\lambda)}{A_{\text{DLER}}(\lambda)} \times 100\,\%. \tag{7}$$

According to Trees et al. (2022), observations with $\Gamma$ smaller than $-15\%$ are most likely affected by cloud shadows. The combined cloud shadow filtering approach consists of filtering out observations for which (1) the PCSF was raised while at the same time (2) the contrast parameter $\Gamma$ is smaller than $-15\%$. This works well, but the complication here is that we have to use the DLER as input for the DLER algorithm itself. This hurdle is by-passed by using a version of the DLER created without cloud shadow filtering. As a result, only the strongest cases of cloud shadow are removed.

For the cloud information needed by the cloud shadow detection algorithm (cloud fraction and cloud height) we use the TROPOMI FRESCO cloud product (Wang et al., 2008, 2012). The settings that were used were taken from Trees et al. (2022). Cloud shadows are removed in almost all cases, at least to the level that they can no longer be detected by eye.

## 4.5 Aerosol screening

For aerosol detection and subsequent filtering of the scene LER observations we make use the absorbing aerosol index (AAI) (Torres et al., 1998; de Graaf et al., 2005; Tilstra et al., 2012) to detect high levels of absorbing aerosols. The AAI product that we use is the official S5P AAI product (Stein Zweers, 2022) and the threshold on the AAI was set to 2 index points. This type of filtering does not remove scattering aerosol, but scattering aerosol increases the scene LER and is therefore removed automatically because the algorithm is looking for the lowest scene LER values to determine the surface LER.

## 4.6 Calculating the surface LER

The traditional, non-directional surface LER database is calculated in the following way. For each calendar month, the observations from all available mission years which are considered cloud-free, clouds shadow-free, and aerosol-free by the screening

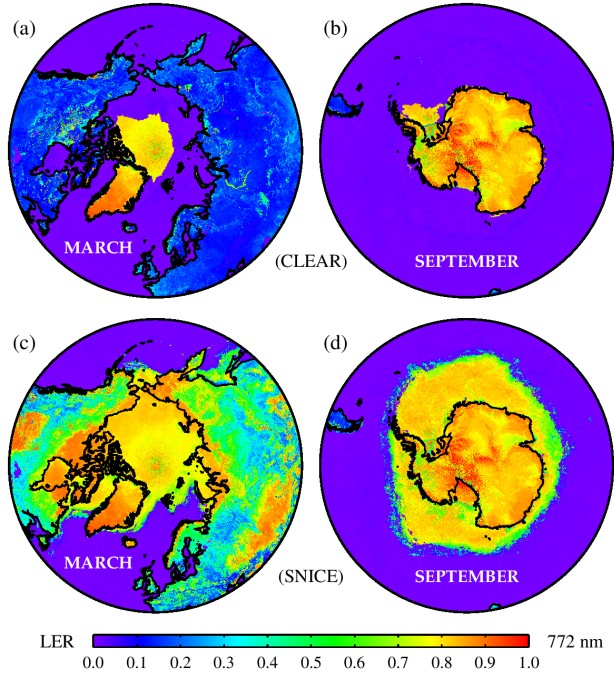

**Figure 1.** Polar maps of the "clear" and "snice" surface LER fields at 772 nm for two different calendar months. To be more specific: (a) March clear field, (b) September clear field, (c) March snice field, (d) September snice field.

steps are mapped onto a 0.125 by 0.125 degrees latitude/longitude grid. In this step, all viewing angles are accepted, although the code can also be instructed to only take a certain viewing angle range into account. The latter possibility is not used here, but it will be used for the DLER calculation introduced in Sect. 4.7. The distribution of the scene LER values of each grid cell is then analysed at a reference wavelength band. This reference wavelength is the longest wavelength in the band duo: 494 nm for band duo 3/4, 772 nm for band duo 5/6 and 2314 nm for band 7 (see Table 1). There are actually two latitude/longitude grids involved: the first grid, called the "clear" grid, only receives observations of snow/ice-free scenes. The second grid only receives observations of scenes which contain snow, permanent ice, or sea ice, and is called the "snice" grid.

For the "clear" grid, the observed scene LER values are sorted at the reference wavelength and the 10% observations having the lowest scene LER values at the reference wavelength are taken apart. From these the average scene LER spectrum is determined, and the result is considered to be the snow/ice-free surface LER spectrum of the grid cell in question. Note that sun glint observations were automatically filtered out because only the 10% observations having the lowest scene LER values were allowed to participate. For the "snice" grid, the histogram of the scene LER distribution is analysed and the histogram bin containing the mode of the distribution is determined. The scenes which fall in the "mode" histogram bin at the reference wavelength are taken apart and the average scene LER spectrum is determined from these. For snow/ice scenes, the mode of the scene LER distribution is considered to be representative for the surface LER (Kleipool et al., 2008; Tilstra et al., 2017). The result is therefore assumed to be the snow/ice surface LER spectrum of the grid cell in question.

Both the "clear" and "snice" grids are incomplete because the requirements w.r.t. the presence or absence of snow/ice cannot always be fulfilled. Gaps are corrected afterwards by copying the missing information from one grid to the other. Figure 1 shows examples of the two grids. The four images show the "clear" and "snice" surface LER grids in the polar regions for two months. Figures 1a and b show the "clear" field for the months March and September, while Figs. 1c and d show the "snice" field for these months. The four figures illustrate that the surface condition extremes (snow or snow-free; ice or ice-free) are contained in the TROPOMI surface DLER climatology. The user of the TROPOMI surface DLER database receives two complete grids to choose from. The "clear" grid is to be used if the user needs snow/ice-free surface albedo, and the "snice" grid is to be used if the user needs surface albedo for snow/ice-presence. In the case of partial snow coverage, the user is advised to mix the "clear" and "snice" values, using the snow cover fraction (if known).

## 4.7 Calculating the DLER

The directional dependence of the surface LER is retrieved in the way best explained by Fig. 2. The figure shows an artificial BRDF, representative for vegetated surfaces. TROPOMI is able to observe such a scene from many different viewing angles over the course of a month. In the retrieval code, the viewing angle range available for this is cut up into nine viewing angle containers and the normal surface LER retrieval introduced in Sect. 4.6 is performed for each of these nine containers. This results in nine surface LER values, which, as a function of viewing angle $\theta_{\mathrm{v}}$, are fitted by a third-order polynomial. The DLER can then be parameterised as a function of the viewing angle $\theta_{\mathrm{v}}$, as was done in Tilstra et al. (2021), however, with a third order term:

$$A_{\mathrm{DLER}} = A_{\mathrm{LER}} + c_0 + c_1 \cdot \theta_{\mathrm{v}} + c_2 \cdot \theta_{\mathrm{v}}^2 + c_3 \cdot \theta_{\mathrm{v}}^3. \tag{8}$$

In Eq. (8), the directional surface LER $A_{\mathrm{DLER}}$ is expressed as an extension on top of the non-directional surface LER $A_{\mathrm{LER}}$. The values of $A_{\mathrm{LER}}$ and of the polynomial coefficients $c_0$, $c_1$, $c_2$, and $c_3$ are contained in the database file. For water surfaces, the polynomial coefficients are set equal to zero. That is, over water surfaces the DLER is identical to the LER and as such represents the diffuse component of the reflection by the water surface (Liu et al., 2020).

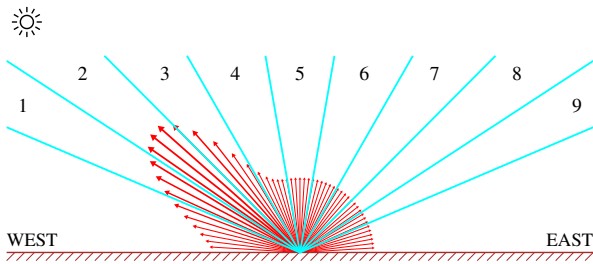

**Figure 2.** Illustration of an artificial BRDF, representative for vegetated surfaces, and the TROPOMI viewing angle range cut up into nine viewing angle containers. A scene is observed multiple times in the course of a month from these nine viewing angle segments.

## 4.8 Post-processing the surface LER and DLER

Several post-processing corrections are conducted to remove imperfections from the surface LER and DLER database. Cloud contamination over the oceans can be detected relatively easily, by checking if the surface LER value exceeds a certain threshold. This is done per band duo, at the longest wavelength band of the band duo. If cloud contamination is detected, then the post-processing correction starts looking for suitable, near-by donor cells. In all cases such a donor cell can be found and the surface LER from the donor cell is copied to the cloud contaminated grid cell. Flags are set to log the situation. This correction is particularly important for the ocean region near 60°S, where the view to the surface is almost always obstructed by clouds.

Contamination by sun glint should not be present at this stage of the processing, because sun glint situations were filtered out quite robustly in the processing step described in Sect. 4.6. If for some reason contamination by sun glint would reach the post-processing step, then this would be detected and treated by the post-processing step in the same way as cloud contamination would be.

Another issue that needs to be addressed is that of missing data due to polar night. Correcting for this phenomenon is mostly a cosmetic procedure, and not relevant for retrievals based on passive instruments. The issue of missing data (empty grid cells) is remedied by the post-processing step by searching for donor cells in other months at exactly the same location. The month nearest in time is used for that.

Other tasks by the post-processing step are performing sanity checking and performing the error calculation. These tasks are described in the ATBD (Tilstra et al., 2023).

## 5 Examples and results

Here, we analyse various properties of the surface LER and DLER database. First, in Sect. 5.1 we study the anisotropy of the Earth's land surface reflectance. In Sect. 5.2 we study the retrieved directional dependence for several surface type regions, as well as its seasonal dependence. In Sect. 5.3 we search for signs of cloud and aerosol contamination in the database. Section 5.4 illustrates how cloud shadows are removed successfully while creating the database.

### 5.1 Surface anisotropy

The directional dependence of the surface reflection can be studied by examining the surface anisotropy parameter. In this paper, this parameter is defined as the difference between the TROPOMI surface DLER at viewing angles $\theta_v$ of $-45°$ (east viewing direction) and $+45°$ (west viewing direction). The results that are shown in Sect. 5.2 confirm that this is a proper definition. In Fig. 3 we have plotted the surface anisotropy parameter for calendar month March and for three wavelength bands: 772, 670, and 463 nm. For 772 nm the anisotropy is quite large, with values reaching 0.16, which corresponds to about 45–80% of the surface DLER at the west viewing direction. These high values are reached for vegetated surfaces and latitudes between 60°S and 60°N. For desert surfaces the anisotropy is much smaller, with values reaching 0.04 or 10–20%. The findings are in agreement with earlier findings reported in a previous paper about surface DLER from the GOME-2 instruments (see

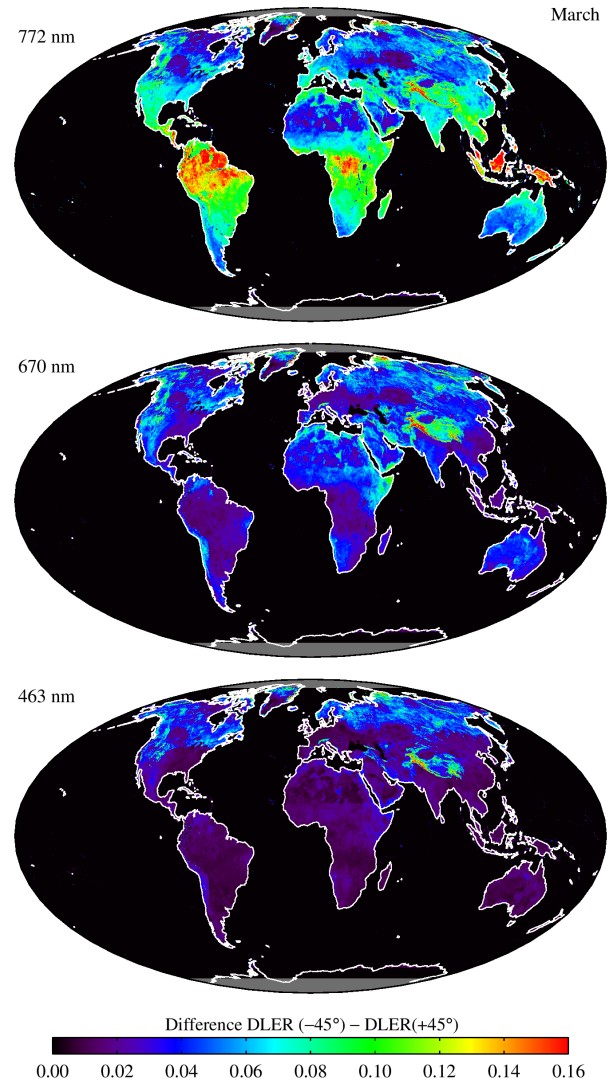

**Figure 3.** Global maps of the surface anisotropy parameter, which is defined as the difference between the TROPOMI surface DLER at viewing angles of −45° and +45°. The results are shown for calendar month March and for three of the wavelength bands. For vegetated surfaces the surface anisotropy parameter can be quite large. This is especially the case for wavelengths beyond 700 nm.

Tilstra et al., 2021, Sect. 6.1). Here surface anisotropy values of 0.2 (vegetation) and 0.05–0.10 (desert) were reported. The values observed by GOME-2 are larger because of the different overpass time of the GOME-2 instruments (09:30 LT), which results in much more asymmetry in the scattering geometries going from east to west across the orbit swath.

For 670 nm, the surface anisotropy parameter derived from the TROPOMI surface DLER database is much smaller than at 772 nm. This is partly because the surface reflectance itself is smaller. For vegetated surfaces the anisotropy parameter now

**Table 2.** Definition of the surface type regions studied in Fig. 4.

| Description | Matthews land type | Latitude range | Longitude range |
| --- | --- | --- | --- |
| Sahara desert | 30 | 16–27°N | 12°W–15°E |
| Arabian Peninsula | 30 | 15–34°N | 37–61°E |
| Australian desert | 30 | 15–30°S | 114–145°E |
| Shrubland | 21 | 36–52°N | 45–114°E |
| Evergreen woodland | 13 | 36–45°N | 10°W–4°E |
| Amazonian tropical rainforests | 1 | 15°S–10°N | 40–85°W |
| Asian (sub-)tropical forests | 2,5,7,9 | 10–35°N | 70–125°E |
| Deciduous forests | 9–11 | 0–40°N | – |
| Grasslands | 23–28 | 35°S–35°N | – |

reaches 0.04 (30–40%); for desert surfaces the numbers are 0.04 (10–20%). At 463 nm, the land surface anisotropy is mild. Values reach about 0.02 for both vegetation and desert, but because of the different reflectance levels the percentages are different (vegetation: 20–40%; desert: ∼10%). Looking at all these numbers and comparing these with previous values found for the GOME-2 surface DLER database, we conclude that the surface anisotropy observed by TROPOMI is smaller than that observed by GOME-2, but still significant. The surface anisotropy will be studied more closely in the next section.

## 5.2 Directional dependence

The anisotropy of the reflectivity of the Earth's surface as observed by the TROPOMI instrument is illustrated in Fig. 4 for a number of land surface types. Table 2 lists the nine land surface type regions that were defined for this purpose. Next to the indicated constraint on the Matthews land usage (Matthews, 1983), restrictions were also set on latitude and longitude. Figure 4 presents the TROPOMI surface DLER as a function of the viewing angle $\theta_v$ defined in Eq. (4) for the nine surface types regions and for four calendar months. The solid curves represent the average surface DLER, which is the mean of the surface DLER values of the participating grid cells. The colour of the curves relates to one of the calendar months as indicated by the legend provided in the "Australian desert" window. The grey curves in Fig. 4 were added to give an idea of the spread in the DLER values. This spread is defined as 2.35 times the standard deviation in the surface DLER data.

The three desert surface regions all show more or less similar behaviour. The dependence on the viewing angle is mild, with variations over the viewing range between 8 and 15% for the "Sahara desert" and the "Arabian Peninsula" regions, and between 20 and 40% for the "Australian desert" region. The average surface DLER is the lowest for the Australian desert. This is most likely caused by the higher levels of vegetation for this region. The temporal behaviour of the Australian desert is different, however, as it is shifted with respect to that of the "Sahara desert" and "Arabian Peninsula" regions. The explanation for this is that the Australian desert is located on the Southern hemisphere. This not only results in a seasonal variation in the surface

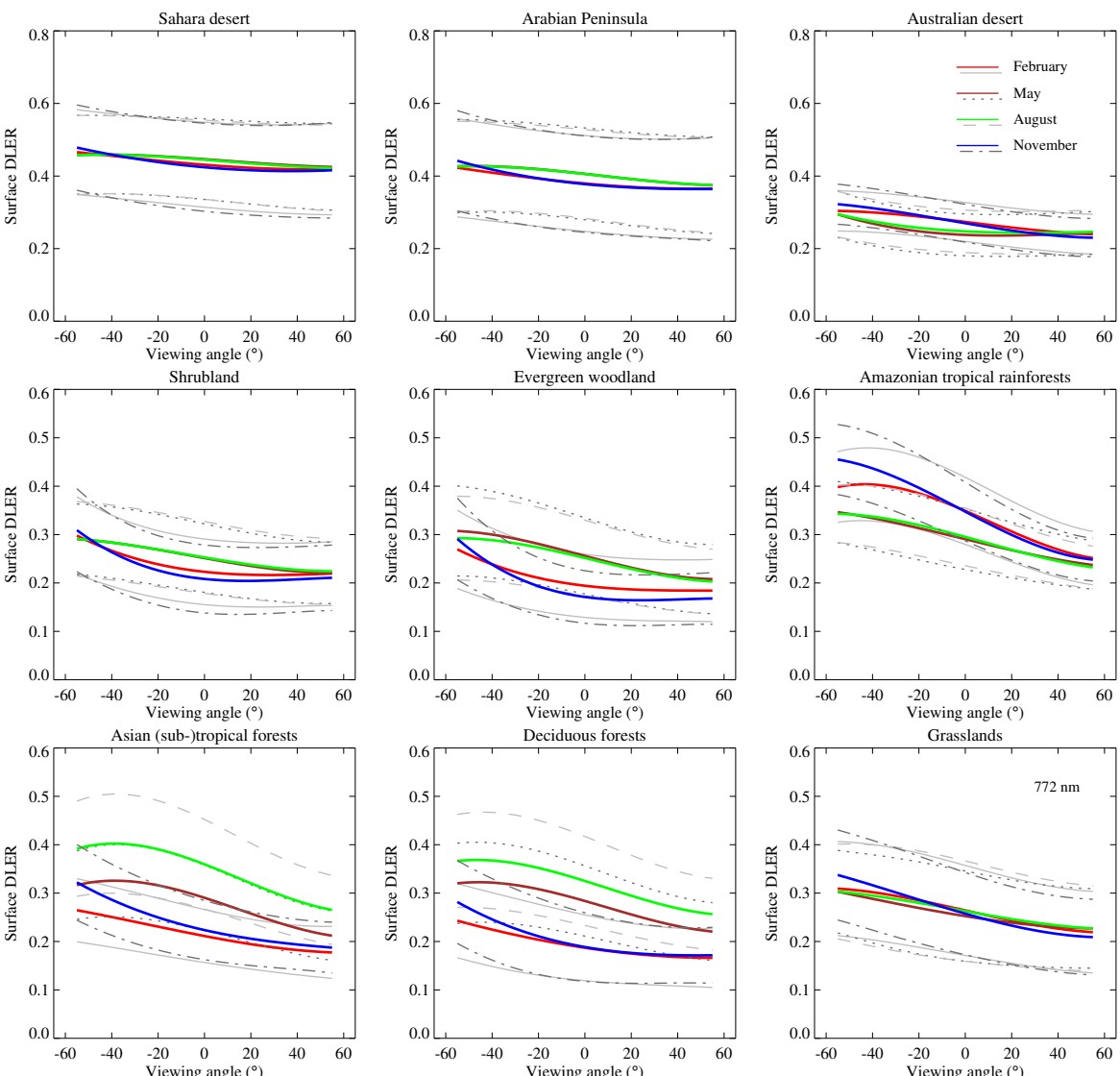

**Figure 4.** TROPOMI surface DLER at 772 nm as a function of the viewing angle $\theta_v$ for the nine land surface type regions defined in Table 2. The coloured curves represent the average surface DLER for the calendar months indicated by the legend (February, May, August, November). The grey curves represent the spread in the surface DLER values, calculated as 2.35 times the standard deviation.

BRDF due to e.g. seasonal changes in vegetation, but also due to the seasonal variation in the solar angles that go into the surface BRDF.

The "Shrubland" and "Evergreen woodland" surface type regions are both located in the Northern Hemisphere. They show similar behaviour, with a mild to strong directional dependence and a fairly strong temporal dependence. The surface albedo values are highest for the months May and August, but the angular variation is strongest for the months February and November.

The "Amazonian tropical rainforests" surface type shows different behaviour than the other vegetated surface type regions. The dependence on viewing angle is quite strong. For instance, for the month November (blue curve) the mean surface DLER varies from 0.25 for $\theta_v$ of $+55°$ to 0.46 for $\theta_v$ of $-55°$. This corresponds to an increase of 83%. Also the temporal behaviour is different than that of the other vegetated surface type regions. The calendar months February and November show higher values and a stronger anisotropy than the calendar months May and August. This is reversed with respect to the other vegetated surface type regions except for the "Grasslands" surface type region. Note that all other vegetated surface types regions except the "Grasslands" one are located in the Northern hemisphere.

The "Asian (sub-)tropical forest" and the "Deciduous forests" surface type regions both show maximum surface DLER in calendar month August. The temporal variation is large. The "Grasslands" surface type region on the other hand shows a very low temporal variation. Similar behaviour was also found earlier from the GOME-2 surface DLER database (Tilstra et al., 2021, Sect. 6.2). In general, the surface anisotropy found for TROPOMI is weaker than that found for GOME-2. This is directly related to the different equator crossing times of the two instruments (TROPOMI: 13:30 LT; GOME-2: 09:30 LT). Results for some of the other wavelength bands can be found in Figs. S1–S3 in the Supplement.

### 5.3 Cloud and aerosol contamination

Despite cloud and aerosol screening and post-processing steps, some degree of cloud and aerosol contamination is unavoidably still left in the final database. Figure 5 presents false colour composite images determined from the retrieved TROPOMI surface LER values, for four calendar months. The images were created using the 402, 494, and 670-nm surface LER intensities serving as the blue, green, and red components, respectively. Cloud contamination would reveal itself by leaving a grey haze in Fig. 5. From earlier analyses performed using previous surface LER databases (see e.g. Koelemeijer et al., 2003; Kleipool et al., 2008; Tilstra et al., 2017) it is known which areas are likely to suffer from cloud contamination. These areas include (i) the northern part of South America, mainly from June till October, (ii) tropical Africa, near the Gulf of Guinea, mainly from January till April, and (iii) certain parts of the ocean.

Indeed, in Fig. 5a one can observe a grey haze in the northern part of South America, near Ecuador. The extent and thickness of the grey haze is smaller than observed for the previous surface LER databases, but it shows that cloud contamination does occur. In the same way, cloud contamination can be observed in Figs. 5b,c near the coast of Guinea. Also in this case the cloud contamination is less strong compared to the previous surface LER databases. This is mainly owing to the smaller footprint size of the TROPOMI instrument, but also the use of active cloud filtering (on top of the statistical approach taken for the previous surface LER databases) may have resulted in less severe residual cloud contamination.

Over the ocean, several examples of cloud contamination can be found, at locations depending on the calendar month. In September (Fig. 5c) a clear grey feature near Angola can be seen. This feature is related to aerosol and cloud presence. The aerosols are biomass burning aerosols which are transported westward over the ocean. Note that the grey feature is also visible

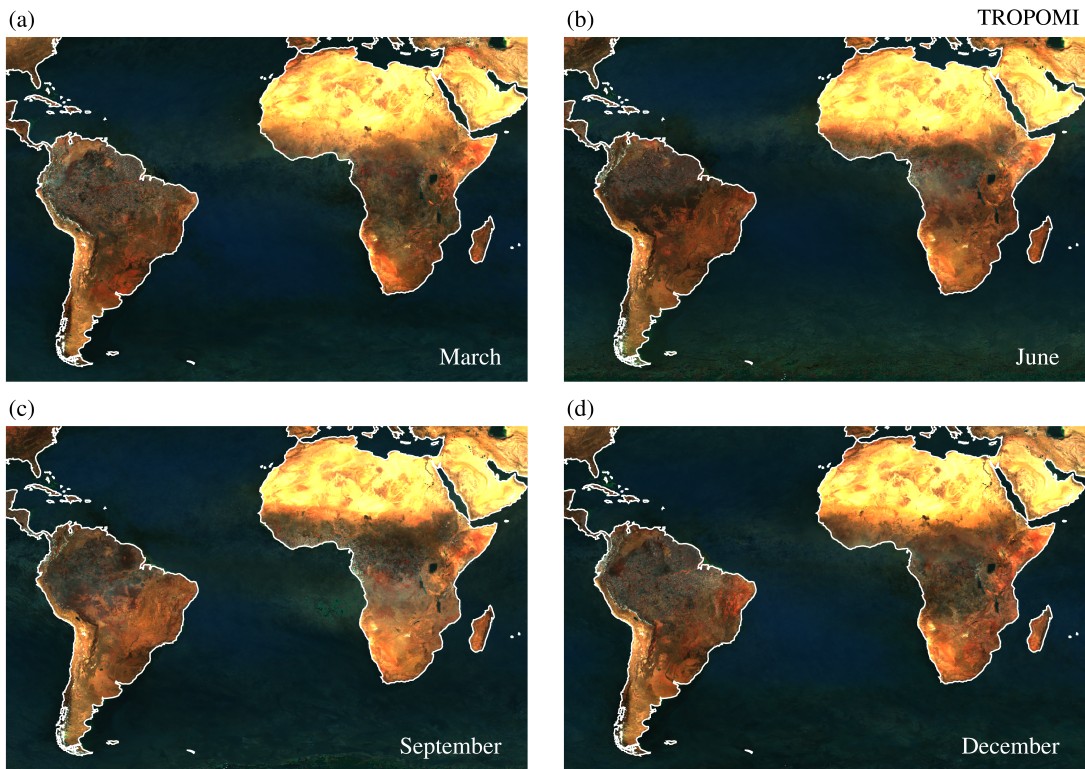

**Figure 5.** False colour composite images, created using the 402, 494, and 670-nm TROPOMI surface LER values, for four calendar months. While producing the images, the surface reflectivity was increased to better emphasise the presence of cloud and aerosol contamination, leading to saturation and discolouration over some of the desert areas. Certain regions show signs of cloud contamination. The impact of persistent aerosol plumes can also been seen in some cases.

over the continent itself, not just over the ocean. In Fig. 5a,b one can also see a grey feature between the north of Africa and the north of South America. These are dust aerosols originating from the desert regions in the north of the African continent and travelling over the Atlantic Ocean. In conclusion, cloud and aerosol contamination are both present in the TROPOMI surface

LER database, although less severe than in previous LER databases.

### 5.4 Impact of cloud shadows

The impact of cloud shadows is not clearly visible in Fig. 5. This is a direct consequence of the cloud shadow screening method introduced in Sect. 4.4. To prove this, Fig. 6 demonstrates the necessity for including cloud shadow filtering in the retrieval. In the left window, a 494-nm "clear" surface LER field is shown which was retrieved from TROPOMI observations from February

2021. The field is based on only one month of data and no post-processing corrections were applied. The field is therefore a very raw field, derived from a low number of observations. This is needed for Fig. 6, however, to be able to emphasise the

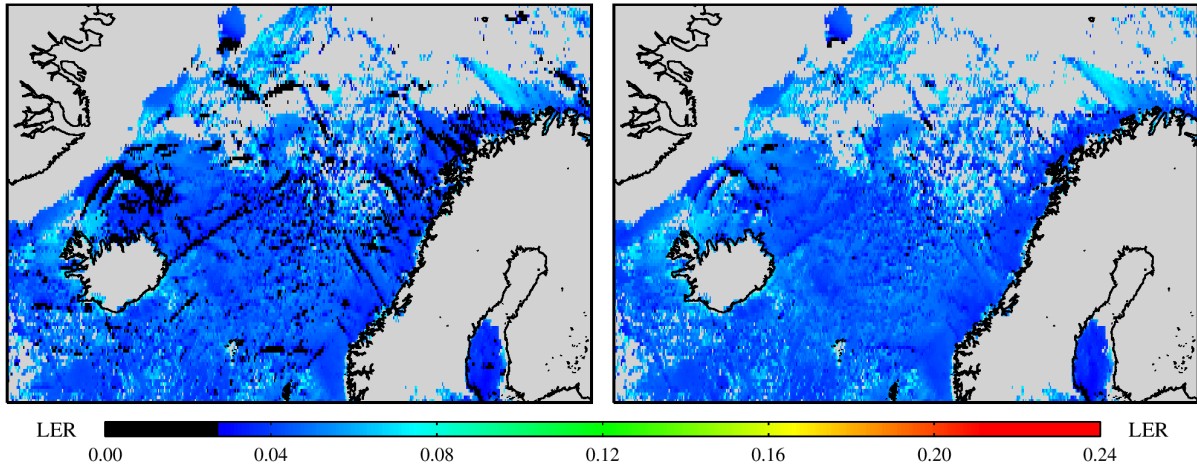

**Figure 6.** Left: The presence of cloud shadows (recognisable as dark stripes) in raw and incomplete surface LER fields of February 2021. The 494-nm "clear" LER field is shown. Right: The same, but now with cloud shadow detection and filtering applied.

impact of cloud shadows. Cloud shadows result in surface LER values lower than usual above the ocean. In the left window, individual cloud shadows can be seen as black features. These cloud shadows are taken into account by the retrieval algorithm which is focused on the lowest scene LER values (as explained in Sect. 4.6). With a low number of observations, the cumulative
mean still contains the signature of individual cloud shadows. Note that because of the large solar zenith angles involved, the dark stripes caused by cloud shadows are relatively thick. In the right window of Fig. 6 the surface LER field was derived from the same data as in the left window, but now with the cloud shadow filtering applied in the retrieval. The black stripes were removed in almost all cases. This means, that the cloud shadow screening is successful.

## 6   Validation

The validation of the (non-directional) TROPOMI surface LER database was based on comparison with several heritage surface LER databases. The validation results, valid for version 2.1 of the DLER database, are presented in Sect. 6.1. The validation of the (directional) TROPOMI surface DLER database was based on comparison with the established MODIS surface BRDF database and on comparison with the OMI GLER database. These validation results, applicable to version 2.1 of the DLER database, are presented in Sect. 6.2 and Sect. 6.3, respectively.

### 6.1   Comparison with heritage LER databases

#### 6.1.1   Approach

In this section we compare the non-directional TROPOMI surface LER with heritage databases based on the GOME-1, OMI, SCIAMACHY, and GOME-2 instruments. These databases are non-directional, except for the GOME-2 DLER database. How-

ever, the DLER expansion of the GOME-2 DLER is valid for the 09:30 LT overpass time of GOME-2, and not for the 13:30 LT
overpass time of TROPOMI. We perform the comparisons for all wavelength bands that can be compared. The results will consist of global maps and histograms of the differences. In the Supplement, tables with detailed validation results will additionally be provided.

### 6.1.2 Results

Figure 7 presents global maps of the differences between the TROPOMI surface LER (v2.1) and the SCIAMACHY surface
LER (v2.6) databases. The differences were determined for the month March and for the four indicated mutual wavelength bands. For the TROPOMI database the snow/ice-free "clear" field was selected; for the SCIAMACHY database the "MIN-LER" field was selected. A grey colour is used for grid cells in the TROPOMI surface LER database which were flagged as corrected for missing data. These are typically grid cells that encountered snow cover during the entire month. From Fig. 7 it can be seen that there is good agreement over the oceans but for the longer wavelengths there are issues over land. For 758 nm,
TROPOMI shows higher values over the desert areas, but lower values over vegetated regions. Here the deviations could be caused by the different overpass times of the TROPOMI and SCIAMACHY satellite instruments, because the surface reflection strongly depends on the solar geometry. For 2314 nm, the differences are harder to explain and surface type alone cannot be

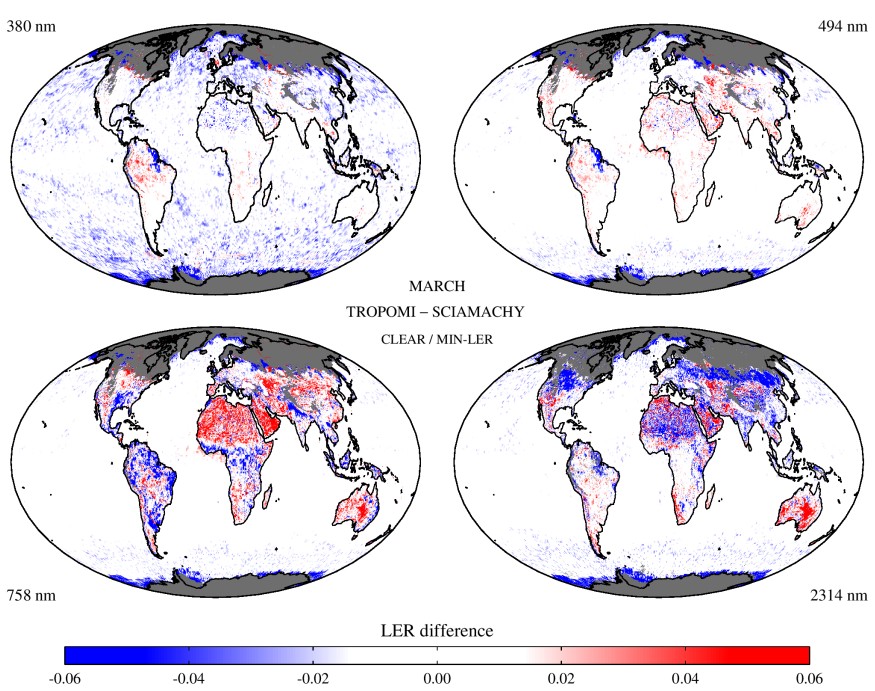

**Figure 7.** Global maps of the differences between the TROPOMI surface LER and the SCIAMACHY surface LER for the month March, for four of the wavelength bands. A grey colour is used for donor grid cells and also for grid cells with snow or ice presence.

the only explanation for the differences because Sahara desert (blue, negative) and Australian desert (red, positive) deviate very differently.

In Fig. 8 the results for the comparison with the OMI surface LER (v3) are shown. Note the different set of wavelength bands compared to Fig. 7. For the shortest wavelength, 380 nm, the differences are similar to the TROPOMI–SCIAMACHY result for 380 nm shown in Fig. 7. This is not surprising as the SCIAMACHY and OMI surface LER databases were found to agree quite well in the past (see e.g. Tilstra et al., 2017, Fig. 12). Nevertheless, the agreement with OMI is clearly better. For the longer wavelength bands, the agreement between TROPOMI and OMI is also good, also for both land and sea surfaces. The

good agreement between TROPOMI and OMI could be expected, because of the orbital similarities and the similar equator passing times. Some mild features are nevertheless visible in Fig. 8.

    Finally, Fig. 9 presents results from the intercomparison between the TROPOMI surface LER (v2.1) database and the GOME-2 surface LER (v4.0) database. There are now some stronger differences visible. At 380 nm, the thick blue haze suggest a systematic difference between TROPOMI and GOME-2 surface LER. The difference, of up to about −0.02, is caused

primarily by a known offset of 0.01–0.02 in the GOME-2 surface LER due to calibration errors (Tilstra et al., 2017). This means that we can conclude that the TROPOMI surface LER should be fairly ok at 380 nm. For the longer wavelengths, there

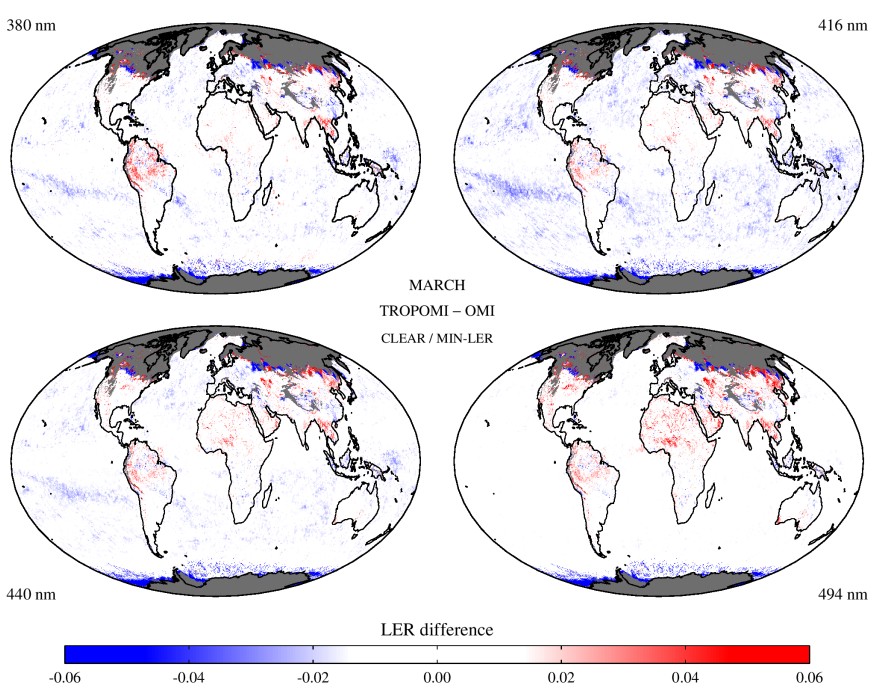

**Figure 8.** Global maps of the differences between the TROPOMI surface LER and the OMI surface LER for the month March, for four of the wavelength bands. For TROPOMI the snow/ice-free "clear" field is used, and for OMI the "MIN-LER" field.

is better agreement between TROPOMI and GOME-2. In fact, compared to the TROPOMI–SCIAMACHY LER comparison in Fig. 7, there are very few differences.

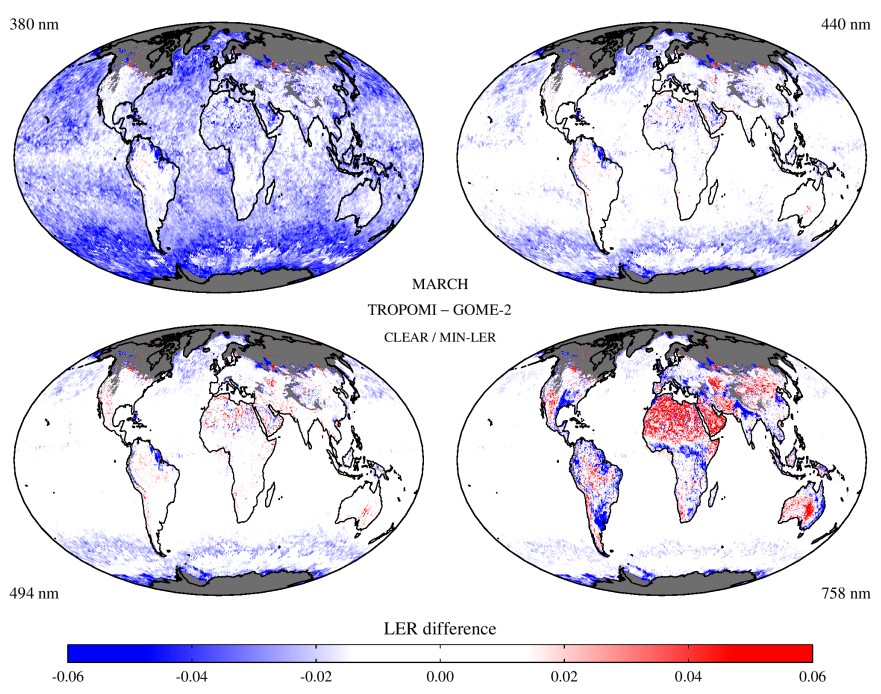

**Figure 9.** Global maps of the differences between the TROPOMI surface LER and the GOME-2 surface LER for the month March, for four of the wavelength bands.

To further study the differences, we determine histograms of the differences. Figure 10 presents a selection of the results.
The histograms were based on the grid cells between 60°S and 60°N in order to skip the polar regions and to avoid variable snow/ice conditions. In Fig. 10a the histogram is that of the difference between the TROPOMI surface LER (v2.1) database and the GOME-1 surface LER (v1.0) database, for 380, 440, 494 and 758 nm. The agreement is rather poor: the distributions are fairly wide and they are shifted with respect to the expected zero difference. It is at this point a bit premature to blame the GOME-1 surface LER database for the differences. Nevertheless, the GOME-1 surface LER database is the most likely
candidate to blame because similar behaviour was seen when this database was compared to the SCIAMACHY, GOME-2, and OMI surface LER databases (Tilstra et al., 2017).

Fig. 10b presents the differences between the TROPOMI surface LER and the GOME-2 surface LER. Here the distributions are less wide and, more importantly, they are becoming less wide with increasing wavelength. This is the expected behaviour, because the surface albedo over water decreases rapidly with increasing wavelength. Additionally, the centre positions of the
415 histograms are close to zero, with the exception of the 380-nm histogram. As reported earlier (Tilstra et al., 2017), there is a

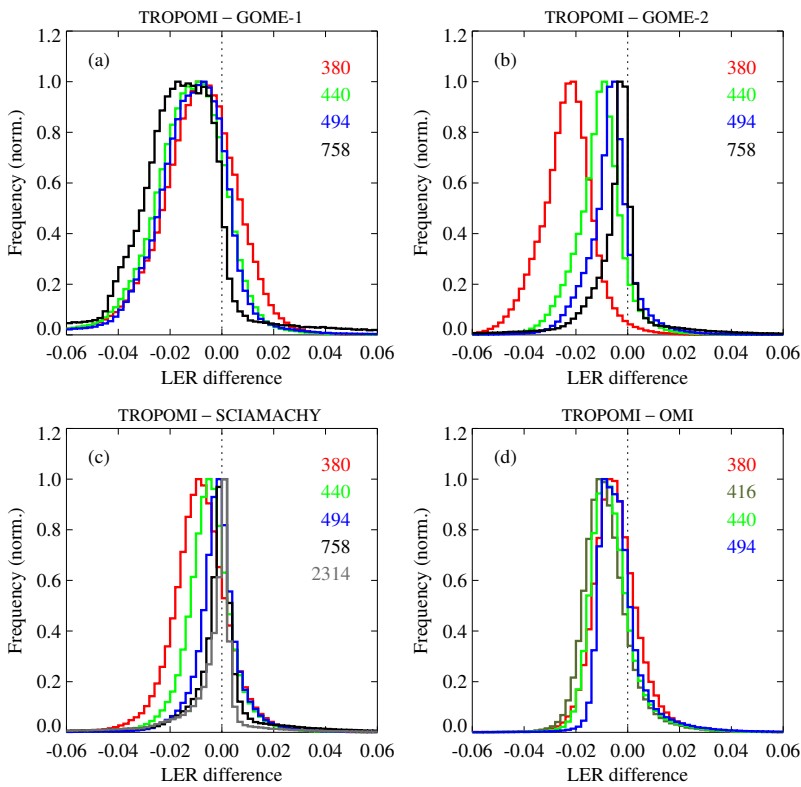

**Figure 10.** Histograms of the differences between the TROPOMI surface LER database and (i) the GOME-1 surface LER database, (b) the GOME-2 surface LER database, (c) the SCIAMACHY surface LER database, and (d) the OMI surface LER database. The histograms were determined for land and water surfaces, for the month of March, for the indicated wavelength bands. To avoid the polar regions, the histograms were based only on grid cells between 60°S and 60°N. In general, there is good agreement.

radiometric calibration error impacting the retrieved GOME-2 surface LER for wavelengths up to 400 nm. This explains the shift of the 380-nm histogram with respect to zero. For the longer wavelength bands, however, there clearly is good agreement.

In Fig. 10c the surface LER difference histograms of the TROPOMI–SCIAMACHY intercomparison are presented. Note that a fifth histogram for the 2314-nm wavelength band was added. The agreement is good for all wavelength bands, with systematic deviations smaller than 0.01 in all cases. Finally, in Fig. 10d the histograms from the TROPOMI–OMI surface LER comparison are shown. The agreement is again good for all wavelengths and the systematic deviations are within 0.01.

For each of the reference databases, histograms were analysed for each calendar month and the central position and width (FHWM) of the distributions were recorded for each wavelength band. In the Supplement, Tables S1–S3 present these results. No dependence on calendar month was found.

## 6.2 Comparison with MODIS BRDF

### 6.2.1 Approach

In this section we compare the TROPOMI surface DLER with MODIS surface BRDF data. Because DLER and BRDF are fundamentally different properties (see Tilstra et al., 2021, Sect. 3.3), we cannot expect to find a perfect agreement between the two. For the UV wavelength range, the Rayleigh optical thickness is high, and there is substantial multiple scattering. In these circumstances there will be quite some light which visits the surface more than once. But, for the longest wavelengths most radiation is only scattered once (and only at the surface). In such cases the DLER and BRDF should be much more alike. We will, therefore, restrict ourselves to the longer wavelengths in the comparisons.

We will compare the TROPOMI DLER and MODIS BRDF databases for the following surface type regions:

1. Libyan desert (25–29°N ; 23–27°E ; February)

2. Sahara desert (16–20°N ; 11–15°E ; February)

3. North America (32–40°N ; 85–100°W ; February)

4. Equatorial Africa (1–7°S ; 17–29°E ; February)

5. Amazonian rainforest (5–15°S ; 55–65°W ; February)

6. Northern Africa (16–29°N ; 8°W–30°E ; February)

7. Australia (21–29°S ; 121–143°E ; February)

8. Greenland (67–78°N ; 35–49°W ; August)

The comparison for cases 1–7 were performed for 15 February 2019. This means that the snow-free MODIS MCD43C2 product from 15 February 2019 was used, and that calendar month February from the TROPOMI DLER database was used. For case 8 the results were obtained for 16 August 2019. For this particular case the snow/ice-containing MODIS MCD43C1 product was used. The more technical details are described in Appendix A.

### 6.2.2 Results

Figure 11 presents a number of representative results from the comparisons between TROPOMI surface DLER and MODIS surface BRDF. To study the performance of the DLER with respect to the LER, also the TROPOMI LER is compared to the MODIS BRDF. In Fig. 11, the first and third column present TROPOMI LER versus MODIS BRDF, for the eight geographical regions defined in Sect. 6.2.1. In a similar way, the second and fourth column in Fig. 11 present TROPOMI DLER against MODIS BRDF. For the Libyan and Sahara desert regions, the optimal wavelength bands are 670 nm for the LER/DLER and 645 nm for the MODIS BRDF. These two wavelength bands are relatively close to each other and long enough to expect a

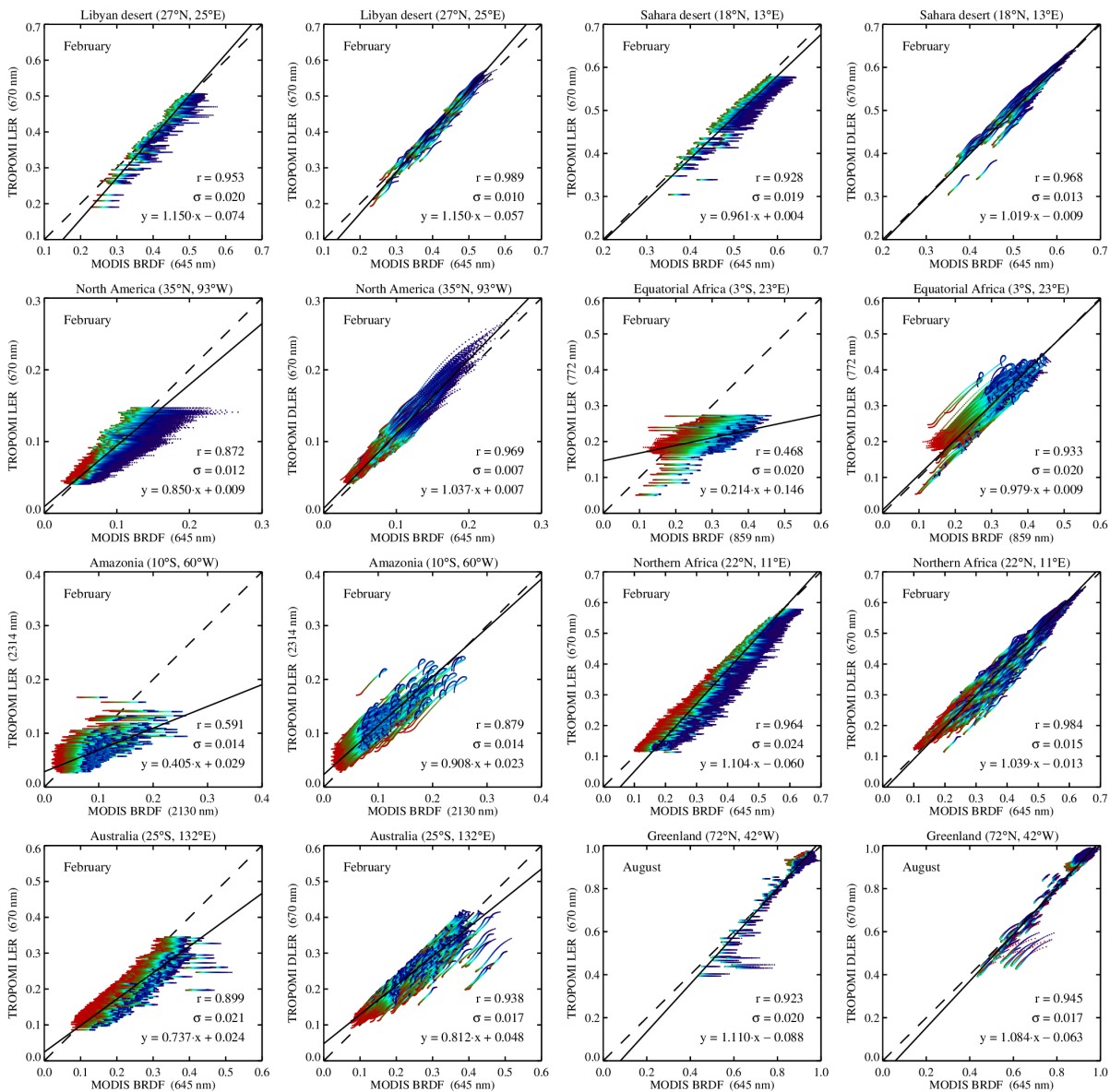

**Figure 11.** TROPOMI surface LER (first and third column) and DLER (second and fourth column) versus MODIS surface BRDF for several geographical regions (indicated by the plot titles). The wavelength bands that were used are indicated in the axes labels. The intercomparison were performed for the month February, except for the Greenland region, for which August was used. In each window, the dashed line indicates the one-to-one relationship and the black line represents a linear fit to the data points. The fit results are discussed in Sect. 6.2.2.

decent agreement between DLER and BRDF. Notice that the LER data points are clustered in horizontal stripes with colours ranging from blue to red, where blue indicates east viewing direction and red indicates west viewing direction. This is because

we simulate the TROPOMI viewing angle range from east to west (see Appendix A), and the LER is non-directional by definition, causing the occurrence of the horizontal stripes.

The improvement in going from LER to DLER is already clear by looking at the data points and the linear fits to the data points (represented by the black solid lines). It can be quantified by calculating Pearson's correlation coefficient ($r$), the standard deviation of the data points with respect to the linear fit ($\sigma$) and the slope and intercept of the linear fit. All these

460 properties are shown in Fig. 11. For the Libyan and Sahara desert, Pearson's $r$ was already pointing to a high correlation for the LER–BRDF comparison but it increases further to strong correlation for the DLER–BRDF comparison. Furthermore, the standard deviation $\sigma$ is reduced considerably, which also points to a better agreement between DLER and BRDF than between LER and BRDF.

The region named "North America" in Fig. 11 is mainly covered by vegetation, which is reflected by the larger viewing angle

dependence. The improvement in going from LER to DLER is therefore quite large. For the region named "Equatorial Africa" we selected different wavelength bands: 772 nm for the LER/DLER and 859 nm for MODIS BRDF. We do this to pass by the vegetation red edge, so that the surface albedo and its anisotropy are larger and therefore better to observe. The downside is that there is a larger mismatch between the two wavelength bands, which reduces the expected agreement. However, the improvement in going from LER to DLER is still quite clear. This is also the case for the Amazonia region. Surfaces with

vegetation benefit the most from the transition from LER to DLER.

To study the performance for snow/ice situations the Greenland region was included. The comparison could only be performed for the months March–September, because of missing data due to polar night in the other months. The results were obtained for the month August. For this month, melting of the Greenland ice sheet causes a larger variation in the surface albedo values, which is good for the analysis that we want to perform. For the Greenland site, both $r$ and $\sigma$ indicate improvement going

from LER to DLER.

## 6.3 Comparison with OMI GLER

### 6.3.1 Approach

In this section we compare the TROPOMI surface DLER database with the OMI GLER database (Qin et al., 2019). Both databases contain directional LER, and the orbits of OMI and TROPOMI are quite similar. The GLER database contains

480 directional LER information for 466 nm. We calculate the TROPOMI LER and DLER at 463 nm using the geographical information and the viewing direction information of the OMI footprints. The GLER data we use are from 15 February 2019.

### 6.3.2 Results

In the top row of Fig. 12 we present maps of the OMI GLER, the TROPOMI LER, and the TROPOMI DLER, for visual comparison. The differences between DLER and LER, LER and GLER, and DLER and GLER, are shown in the bottom row.

From the top row it can be seen that, on the whole, GLER and LER agree fairly, but that there are also clear differences. For instance, the reddish feature over the Egyptian desert is less pronounced in the LER than in the GLER. The DLER compares

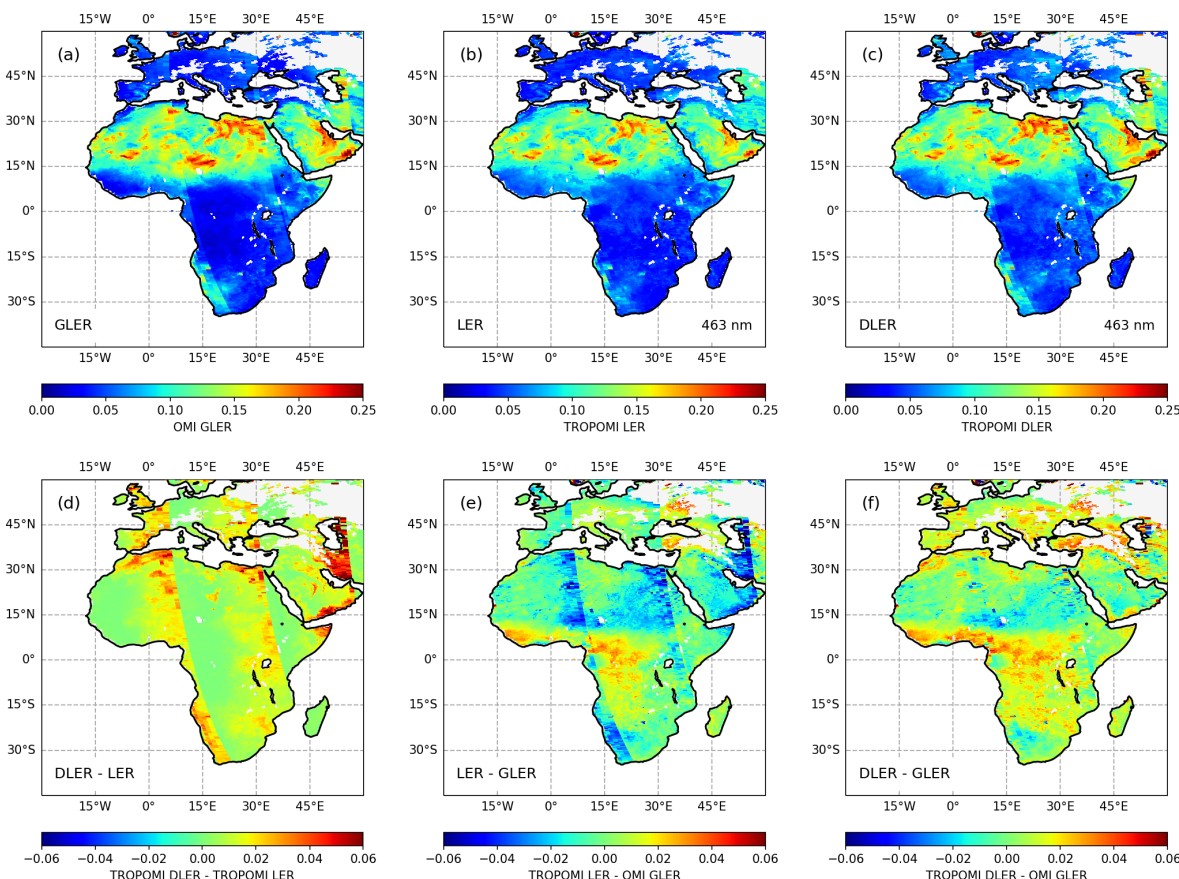

**Figure 12.** Maps of (a) OMI GLER, (b) TROPOMI LER, (c) TROPOMI DLER, (d) the difference between DLER and LER, (e) the difference between LER and GLER, and (f) the difference between DLER and GLER. The OMI orbits that were used as a basis for the comparison are from 15 February 2019. The 463-nm DLER wavelength band was used for the comparison.

much better with the GLER for this particular case, but also for a number of other cases, for instance in the region below the Caspian Sea and the region at the coasts of Namibia and South Africa.

The bottom row of Fig. 12 shows the magnitude of the differences. Figure 12d shows the difference between DLER and LER. The individual OMI orbit swaths are clearly visible. Note that this merely reveals the fact that we are calculating the DLER and LER for individual OMI orbits, using also the viewing direction information. The DLER and GLER by definition depend on the viewing direction, but the LER does not. From Fig. 12d it can also be seen that DLER differs from LER mostly on the west side of the orbit swath. Looking at Fig. 12e we see that the LER deviates most clearly from the GLER at the west side of the orbit swath. The difference between DLER and GLER in Fig. 12f does not show this particular behaviour. In general, differences between DLER and GLER are smaller than 0.02 but slightly larger differences are also observed.

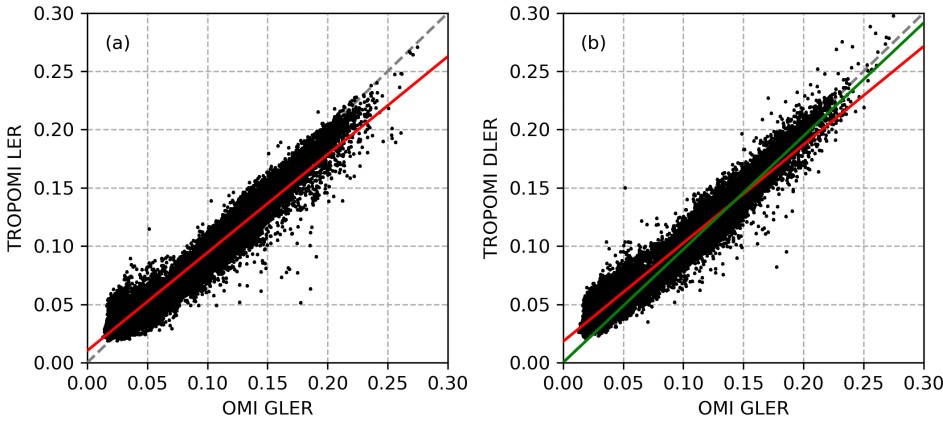

**Figure 13.** Left: TROPOMI LER versus OMI GLER. Right: TROPOMI DLER versus OMI GLER. The data points were calculated for OMI orbits 77592 and 77593 from 15 February 2019. The red lines represent linear fits to the data points. The green line in the right window is also a linear fit, but only applied to the data points for which TROPOMI DLER and OMI GLER are both larger than 0.07.

To further study the differences, Fig. 13 presents scatter plots of (a) TROPOMI LER versus OMI GLER and (b) TROPOMI DLER versus OMI GLER. The data points were calculated for OMI orbits 77592 and 77593, both from 15 February 2019. The red lines represent linear fits to the data points. In both cases (a) and (b), the Pearson correlation coefficient is on the order of 0.98 and the standard deviation of the data points w.r.t. the linear fit is ∼0.01, suggesting a good linear correlation. However, the linear fits in both cases deviate somewhat from the expected one-to-one relationship. The deviation seems to be caused mainly by data points representing low surface reflectivity. If we perform the linear fit only to data points with DLER and GLER larger than 0.07 (green line in Fig. 13b) then the linear fit is very close to the one-to-one relationship.

Positive offsets like in Fig. 13 and patterns like in Fig. 12 have been reported earlier when the OMI surface LER database was compared to the MODIS black-sky albedo (BSA) database (Kleipool et al., 2008, Fig. 7). Since the OMI GLER database effectively uses MODIS BRDF as input, one explanation for the differences that we see might be calibration issues that affect the MODIS BRDF product. However, there are many other possible explanations, so this remains speculation. The conclusion that can be drawn is that there is a good correlation between DLER and GLER, and that the DLER is closer to the GLER than the LER due to the fact that the DLER is directional, while the LER is not.

## 6.4 Discussion of results

In general, there is good agreement between the TROPOMI surface LER and the heritage surface LER databases based on GOME-1, OMI, SCIAMACHY and GOME-2. From the global difference maps presented in Figs. 7–9 we concluded that the best agreement is found when the TROPOMI surface LER is compared with the OMI surface LER. This can be understood by looking at the orbital and instrumental similarities of the two satellite instruments. The TROPOMI and OMI instruments have similar designs and their orbits have comparable local equator crossing times (13:30 and 13:45 LT, respectively). The similarity

in overpass time means that the solar geometries are more or less the same. This means that the viewing and solar angles that define the surface BRDF are similar, resulting in a reliable comparison. But also the agreement with the SCIAMACHY and GOME-2 surface LER databases is satisfactory.

More quantitative results, derived from histograms such as the ones presented in Fig. 10, were derived for all calendar months. For the wavelength bands from 328 to 494 nm we find accuracies within 0.01–0.02 when we compare with the OMI and SCIAMACHY surface LER databases. This is well within the accuracy requirements of 0.03+10%. Comparing with the GOME-2 surface LER database, we find an offset of about −0.02 below 400 nm, which we can attribute to an offset in the GOME-2 surface LER database that was already reported earlier (Tilstra et al., 2017). For the longer wavelengths (670–2314 nm) we found biases below 0.003, which is better and well within the accuracy requirements of 0.02+10%. A dependence on calendar month was not found. Detailed results can be found in the Supplement (Figs. S4–S6 and Tables S1–S3).

Good agreement between TROPOMI surface DLER and MODIS surface BRDF was found for a range of surface type regions. In all cases, the DLER performed significantly better than the LER. The comparison was in most cases performed for the DLER wavelength band at 670 nm and the MODIS band at 645 nm. This is the only combination of wavelength bands which can be used for a quantitative analysis, so the conclusion of good agreement in principle only applies to the 670 nm wavelength band. However, combined with the qualitative results from longer wavelength bands we expect that the angular dependence is correct for all wavelength bands.

The comparison showed that there is a good correlation between DLER and BRDF, with correlation coefficients close to one. The slopes of the linear fits were in most cases also close to one, but it is not easy to draw conclusions from them for various reasons. One reason is the already mentioned mismatch between wavelength bands of TROPOMI DLER and MODIS BRDF. Another factor may be the existence of radiometric calibration errors. Calibration errors were found in the past for TROPOMI spectral band 3 and 4, with reported errors between 6% and 10% (Tilstra et al., 2020). However, these numbers apply to an older version of the TROPOMI level-1 data (v1.0.0). The calibration should have been improved significantly for the version of the level-1 data that were used for the creation of the DLER database (v2.1.0). Moreover, the DLER wavelength bands that were compared with MODIS BRDF were derived from TROPOMI spectral bands 5–7, not from spectral bands 3 or 4.

In the comparison the MODIS BRDF from one day was compared with the DLER representative for the entire month. This is not optimal and can cause differences. Another factor to take into account is the fact that DLER and BRDF are fundamentally different properties (cf. Sect. 6.2.1). There are, in practice, many factors that prevent a perfect comparison between TROPOMI DLER and MODIS BRDF. However, the goal of the comparison was not to perform a very reliable validation. The comparison with MODIS BRDF does clearly show that the DLER is a considerable improvement on the LER. In the Supplement, more detailed results can be found in Figs. S7–S15.

The TROPOMI surface DLER database was also compared with the OMI GLER database around 466 nm. The results clearly show that the DLER is closer to the GLER than the LER is, as expected. Data analyses showed good correlation, with Pearson's correlation coefficient around 0.98. However, there seems to be a positive bias in the DLER values for low values of the surface albedo. For surface albedo values above $\sim 0.07$, this bias seems to be absent.

## 7 Conclusions

This paper introduced a new surface albedo climatology of directionally dependent Lambertian-equivalent reflectivity (DLER) observed by the TROPOMI instrument on the Sentinel-5 Precursor satellite. The database contains monthly fields of DLER for 21 wavelength bands at a relatively high spatial resolution of $0.125° \times 0.125°$. The anisotropy of the surface reflection is handled by parameterisation of the viewing angle dependence.

  The surface anisotropy and the seasonal cycle captured by the DLER were studied for various surface types. The behaviour 555 was found to be similar to behaviour also found earlier in the GOME-2 surface DLER climatology. Differences in behaviour were also found but these could be related to the difference in solar position due to difference in the overpass times of GOME-2 and TROPOMI.

  The TROPOMI surface DLER database was validated first by intercomparison of the non-directional surface LER with traditional surface LER databases based on GOME-1, OMI, SCIAMACHY and GOME-2. The comparisons with OMI and 560 SCIAMACHY showed good qualitative and quantitative agreement for all wavelength bands and for all calendar months. The accuracies were found to be below 0.01–0.02 for the shorter wavelengths (328–494 nm). These accuracies are below the requirements that were set (0.03+10%). For the longer wavelengths (670–2314 nm) we found biases below 0.003. These accuracies are well below the accuracy requirements (0.02+10%). The comparison with GOME-2 showed good agreement for the longer wavelengths. For the shorter wavelengths, biases were found which could be linked to systematic biases in the 565 GOME-2 surface LER database for these wavelengths. These systematic biases for the GOME-2 surface LER database were reported earlier (Tilstra et al., 2017). For the TROPOMI surface LER database, the overall conclusion is that all requirements are met.

  Next, the TROPOMI surface DLER database was compared with MODIS surface BRDF for a collection of scenes. The non-directional TROPOMI LER, which is also available in the DLER database, was also compared with MODIS BRDF to 570 study the improvement in going from LER to DLER. The LER shows fair to reasonable correlation with MODIS BRDF, but the DLER is clearly better and has good to very good correlation with MODIS BRDF. The comparison was performed for various wavelength bands, but only the results from the comparison at 670 nm can be interpreted in a fairly quantitative way. Even for this wavelength band there are complications preventing a quantitative comparison, such as the difference in wavelength between the TROPOMI wavelength band used (670 nm) and the MODIS BRDF wavelength band used (645 nm). 575 Nevertheless, the numbers that are found point to a good correlation between TROPOMI DLER and MODIS BRDF, and the conclusion is that the DLER is an important improvement on the traditional, non-directional LER. Detailed information about the results is provided in the Supplement.

  Finally, the TROPOMI surface DLER database was compared with the OMI GLER database for a land surface area covering Africa, Europe and the Middle East. Good correlation is found between DLER and GLER. A small bias is found for scenes 580 with a low value of the surface albedo. Again it is found that the DLER performs better than the LER in the sense that the DLER is generally speaking closer to the OMI GLER than the LER is.

The TROPOMI surface DLER database is being used in the official TROPOMI data processing as input for various level-2 products, including the ozone profile retrieval, the retrieval of nitrogen dioxide, the FRESCO cloud product retrieval, and the retrievals of aerosol layer height and aerosol optical thickness. The TROPOMI surface DLER database can also be used for the support of retrievals from observations made by the OMI instrument, because the overpass times of TROPOMI and OMI are very similar. The DLER algorithm setup that was presented in this paper can be adapted to retrieve DLER from other instruments as well, for instance, from Sentinel-5 and 3MI on the MetOp-SG-A1 satellite to be launched in 2025, from OLCI on the Sentinel-3 satellites, and from the future CO2M mission.

*Data availability.* The TROPOMI surface DLER database can be downloaded from the TEMIS website via the following URL: https://www.temis.nl/surface/albedo/tropomi_ler.php.

## Appendix A: Definition of viewing and solar geometries

To be able to compare TROPOMI surface DLER and MODIS surface BRDF we first need to define the viewing and solar angles that are involved. We start out with an artificial array of (signed) viewing angles $\theta_v$ which range between $-66.3°$ and $+66.3°$ in 101 steps, where the minus sign indicates an east viewing direction and the plus sign indicates a west viewing direction. Already this definition of $\theta_v$ is enough to calculate the TROPOMI DLER for any of the grid cells inside the TROPOMI surface DLER field.

Finding the complete set of angles needed for calculating MODIS surface BRDF ($\theta, \theta_0, \phi - \phi_0$) is a bit more complicated. For every artificial $\theta_v$ of the grid cell at hand we determine the associated solar zenith angle $\theta_0$ and relative azimuth angle $\phi - \phi_0$. This is done on the basis of the TROPOMI viewing and solar geometry from a TROPOMI orbit of the same day, using only $\theta_v$ and the central latitude of the grid cell as input. The viewing zenith angle $\theta$ is by definition the absolute value of $\theta_v$. In other words: $\theta = |\theta_v|$. With all three angles known, the kernels $K_{vol}$ and $K_{geo}$ can be calculated using the equations provided in Appendix B. The kernel coefficients $f_{iso}$, $f_{vol}$, $f_{geo}$ are then determined from the MODIS MCD43C1/2 product of the given day. After that, the MODIS surface BRDF can be calculated according to:

$$
\begin{aligned}
A_g(\lambda, \theta, \theta_0, \phi - \phi_0) = f_{iso}(\lambda) \\
+ f_{vol}(\lambda) \cdot K_{vol}(\theta, \theta_0, \phi - \phi_0) \\
+ f_{geo}(\lambda) \cdot K_{geo}(\theta, \theta_0, \phi - \phi_0)
\end{aligned}
\tag{A1}
$$

The TROPOMI DLER and MODIS BRDF can then be compared for the many grid cells in the geographical regions that were defined in Sect. 6.2.1. More precisely, this step involves binning of the smaller MODIS BRDF grid cells ($0.05° \times 0.05°$) and the larger TROPOMI DLER grid cells ($0.125° \times 0.125°$) to common grid cells of $0.25° \times 0.25°$.

## Appendix B: Kernels for the Ross–Li BRDF model

This appendix lists the equations needed to calculate the kernels that make up the Ross–Li BRDF model of surface reflectance. Proper derivations of the Ross–Thick and Li–Sparse kernels can be found in Wanner et al. (1995).

### B1  Ross–Thick volumetric kernel

The Ross–Thick volumetric scattering kernel is defined in the following way (Roujean et al., 1992):

$$K_{\text{vol}} = \frac{(\pi/2 - \xi)\cos\xi + \sin\xi}{\cos\theta + \cos\theta_0} - \frac{\pi}{4} . \tag{B1}$$

In Eq. (B1), $\theta$ refers to the viewing zenith angle and $\theta_0$ to the solar zenith angle. The angle $\xi$ is defined according to

$$\cos\xi = \cos\theta\cos\theta_0 + \sin\theta\sin\theta_0\cos(\psi - \psi') , \tag{B2}$$

where $\psi$ and $\psi'$ are the viewing and solar azimuth angles following the definition in Strahler et al. (1999). Exact backscattering ($\xi = 0°$) occurs for $\psi - \psi' = 0°$, which agrees with the definition used for the GOME-2 data products.

### B2  Li–Sparse geometric kernel

The Li–Sparse geometric scattering kernel (Li and Strahler, 1986) is defined as:

$$K_{\text{geo}} = O - \sec\theta^\star - \sec\theta_0^\star + \frac{1}{2}(1 + \cos\xi^\star)\sec\theta^\star\sec\theta_0^\star . \tag{B3}$$

The term $O$ in Eq. (B3) and the starred angles $\theta^\star$, $\theta_0^\star$, and $\xi^\star$ are calculated using the following set of equations:

$$\theta^\star = \arctan\left(\frac{b}{r}\tan\theta\right) , \quad \theta_0^\star = \arctan\left(\frac{b}{r}\tan\theta_0\right) , \tag{B4}$$

$$\cos\xi^\star = \cos\theta^\star\cos\theta_0^\star + \sin\theta^\star\sin\theta_0^\star\cos(\psi - \psi') , \tag{B5}$$

$$O = \frac{1}{\pi}(t - \sin t\cos t)(\sec\theta^\star + \sec\theta_0^\star) , \tag{B6}$$

$$\cos t = \frac{h}{b}\frac{\sqrt{D^2 + (\tan\theta^\star\tan\theta_0^\star\sin(\psi - \psi_0))^2}}{\sec\theta^\star + \sec\theta_0^\star} , \tag{B7}$$

$$D = \sqrt{\tan^2\theta^\star + \tan^2\theta_0^\star - 2\tan\theta^\star\tan\theta_0^\star\cos(\psi - \psi')} . \tag{B8}$$

The parameters $b/r$ and $h/b$ are the crown relative shape and the crown relative height, respectively. These were fixed to 1 and 2, respectively, following Strahler et al. (1999).

*Author contributions.* LGT wrote the manuscript, developed the algorithm, and performed the validation. MdG helped with aerosol retrieval and filtering and supported the development. VJHT contributed the cloud shadow detection algorithm. PL and OD advised during the development phase of the project. PS and MdG helped with the radiative transfer modelling. All authors discussed the results and commented on the manuscript.

*Competing interests.* At least one of the (co-)authors is a member of the editorial board of Atmospheric Measurement Techniques.

*Acknowledgements.* The work that was presented in this paper was supported by ESA via the Sentinel-5p+ Innovation project. ESA is also
acknowledged for providing the TROPOMI data. Data processing was performed primarily on the Sentinel-5P Product Algorithm Laboratory
(S5P-PAL) developed and hosted by S[&]T. We thank the two anonymous reviewers for their constructive comments.

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
