# Peer review of "A directional surface reflectance climatology determined from TROPOMI observations"

_Atmospheric Measurement Techniques, 2023_

## Author Comment (AC1)

**Response to RC2 by Reviewer 2:**

We would like to thank the reviewer for performing a thorough review and for the many helpful suggestions to improve our manuscript.

Below, we respond to each of the review comments. For the sake of clarity, the review comments are given in blue italics and our response is printed in normal font. Changes to the manuscript are printed in green.

*The paper describes an important and timely step that should substantially improve characterization of the Earth's surface reflectivity derived from the TROPOMI observations. Following Vasilkov et al. (2017 - OMI), Loyola et al. (2020 - TROPOMI) and Tilstra et al. (2021 - GOME-2), the paper re-introduces the directionally dependent Lambertian-equivalent reflectivity (DLER) for TROPOMI and compares the results with various non-directional LER data sets. Such comparisons are indicative, though not conclusive, considering the profound differences in the LER and DLER approaches (e.g., the viewing-angle DLER dependencies from Fig. 4, cf. the angle-independent LER). Hence, the authors validate the results by comparing TROPOMI DLERs to the MODIS surface BRDFs, however only in the near-IR region. The $\lambda < 500$ nm domain is critically important for retrievals of many trace-gas species. Surprisingly, the authors completely avoid any comparisons with the publicly available database that is built on similar (directionally-dependent LER, known as GLER: Vasilkov et al. 2017) physical principles, samples the Earth's surface at similar local solar times, and belongs to the wavelength region of interest:*

*https://disc.gsfc.nasa.gov/datasets/OMGLER_003/summary?keywords=AURAMLS*

*This can be perceived as a serious deficiency of the reviewed study unless there are sound reasons for excluding these data from consideration. If any, such reasons should be explicitly stated and commented on.*

Thank you very much for bringing this to our attention. We were well aware of the existence of the GLER database, but we did not know that the database had become publicly available. The GLER database is indeed well suited as a reference, because the OMI and TROPOMI orbits are similar, meaning that we end up with comparable viewing and solar geometries. We have, therefore, downloaded the necessary GLER data and have set up a comparison between the OMI GLER database and the TROPOMI DLER database.

Some of the results are shown in Figures 1 and 2 of this AC. In the top row of Figure 1 we show maps of the OMI GLER, the TROPOMI LER, and the TROPOMI DLER, for visual comparison. The differences between DLER and LER, LER and GLER, and DLER and GLER, are shown in the bottom row. From the top row it is clear that, roughly speaking, there is agreement between GLER and DLER, but that there are also differences between GLER and DLER. The bottom row shows the magnitude of the differences. In general, differences are smaller than 0.02 but larger differences are also found.

In Figure 2 of this AC, scatter plots are shown of (a) TROPOMI LER versus OMI GLER and (b) TROPOMI DLER versus OMI GLER. The data points originate from OMI orbits 77592 and 77593, both from 15 February 2019. The red lines represent linear fits to the data points. In both cases (a) and (b) the Pearson correlation coefficient is on the order of 0.98 and the standard deviation of the data points w.r.t. the linear fit is ∼0.01, suggesting a good linear correlation. However, the linear fits in both cases deviate somewhat from the expected one-to-one relationship. The deviation seems to be caused mainly by data points representing low surface

[Figure]

Figure 1: Maps of (a) OMI GLER, (b) TROPOMI LER, (c) TROPOMI DLER, (d) the difference between DLER and LER, (e) the difference between LER and GLER, and (f) the difference between DLER and GLER. The OMI orbits that were used as a basis for the comparison are from 15 February 2019. The 463-nm DLER wavelength band was used for the comparison.

[Figure]

Figure 2: Left: TROPOMI LER versus OMI GLER. Right: TROPOMI DLER versus OMI GLER. The red lines represent linear fits to the data points. The green line in the right window is also a linear fit, but only applied to the data points for which TROPOMI DLER and OMI GLER are both larger than 0.07.

reflectivity. If we perform the linear fit only to data points with DLER and GLER larger than 0.07 (green line in Figure 2b) then the linear fit is very close to the one-to-one relationship.

Positive offsets like in Figure 2 and patterns like in Figure 1 have been reported earlier when the OMI surface LER database was compared to the MODIS black-sky albedo (BSA) database [Kleipool et al., 2008, Fig. 7]. Since the OMI GLER database effectively uses MODIS BRDF as a basis, one explanation for the differences that we see might be calibration issues that affect the MODIS BRDF product. However, there are many other possible explanations, so this remains speculation.

The reviewer is thanked again for bringing the OMI GLER to our attention and the results and discussion, including Figures 1 and 2 of this AC, have been included in the revised manuscript, in the new section 6.3.

*More comments:*

*In Section 2 (Description of TROPOMI) the authors should mention what L1B version is used in the study. This should be tied to the information on how the instrument degradation is accounted for, if applicable to the case. Moreover, the very important and relevant topic of the quality of absolute TROPOMI calibration should be described in some detail. If pertinent, the latter may explain some systematic differences shown in Section 6. Can the introduced $c_0$ term (Eq. 8) be explored/exploited as a potential link to the absolute calibration uncertainties?*

We have added the L1B version number to the introduction of the manuscript. A correction for instrument degradation is an integral part of the latest version (v2.1.0) of the L1B product, so we do not perform our own degradation correction. Radiometric calibration errors did exist in spectral bands 3 and 4 for earlier versions of the TROPOMI L1B data, however, these issues have been resolved and the latest version (v2.1.0) of the L1B product is well calibrated in the absolute sense. The quality of the absolute calibration of TROPOMI cannot, of course, be described in full detail in the manuscript but several references exist, to which we now refer in section 2 of the manuscript.

The text in section 2 of the manuscript now reads:

" The radiometric calibration of the TROPOMI instrument has been improved a number of times since its launch. The latest version of the level-1b data (v2.1.0) includes, amongst other things, a correction for instrument degradation. An issue in the radiometric calibration of spectral bands 3 and 4 [Tilstra et al., 2020] has been resolved in this version. More information about the TROPOMI instrument, its calibration, and the products derived from it can be found in Kleipool et al. [2018]; Ludewig et al. [2020] and in Veefkind et al. [2012]. "

*l.173 – briefly comment on the 'undisputed cases.' Is/are there any threshold value(s) bounding such cases?*

The geometrical cloud fraction as defined in equation (6) is only based on the fraction of confidently cloudy VIIRS observations. The fraction of probably cloudy VIIRS observations is not taken into account, making the geometrical cloud fraction in equation (6) less strict. Basically, the geometrical cloud fraction only responds to confidently cloudy VIIRS observations, to which we refer in the text as "undisputed cases of cloud cover". The threshold value that we use for the filtering is the value 0.03 mentioned in the manuscript.

*l.199 – mention the source of the AAI data used for screening.*

We have added the proper reference to the TROPOMI AAI product:

" . . . The AAI product that we use is the official S5P AAI product [Stein Zweers, 2022] and the threshold on the AAI was set to 2 index points. . . . "

*l.217 – the mode of a distribution could be a robust metric for the scenes with permanent ice/full snow coverage. However, this may not be applicable to the partial/thin-snow landscapes. Is there any recipe for addressing such cases that, if mis-represented, may profoundly bias the trace-gas retrievals? Can (should) the 'clear' and 'snice' fields be mixed in the cases of partial coverages? Please share your experiences (know-how) with this sensitive scenario.*

Yes, this is true. Grid cells with permanent snow/ice coverage are retrieved better (and probably perform better) than grid cells filled with scenes that are only partially covered by snow/ice. The "clear" and "snice" fields are supposed to represent the extreme cases of "definitely no snow/ice present" and "definitely snow/ice present". For situations in which scenes are only partially covered by snow/ice, we would indeed advise to calculate the surface albedo from the clear and snice fields, using the snow/ice cover fraction (if known).

The last paragraph of section 4.6 of the manuscript now reads:

" . . . The "clear" grid is to be used if the user needs snow/ice-free surface albedo, and the "snice" grid is to be used if the user needs surface albedo for snow/ice-presence. In the case of partial snow coverage, the user is advised to mix the "clear" and "snice" values, using the snow cover fraction (if known). "

*l.239 (also applicable to Section 4.8, i.e., the post-processing routines) – please describe how the solar-glint regions are incorporated (filtered? outright rejected?) under the approach.*

Sun glint is not treated differently by the retrieval code. For the "clear" field, the algorithm is focused on the lowest scene LER observations (as explained in section 4.6). Sun glint observations are therefore filtered out automatically because sun glint scenes have a higher reflectance, like clouds have. This method works well. The retrieval code does have the possibility to exclude sun glint observations, because these are flagged in the TROPOMI L1 and L2 products. However, we decided not use this option. For the "snice" field, sun glint is not an issue, because the ice-free ocean is skipped in this retrieval mode.

Sun glint should indeed have been mentioned. The text in section 4.6 of the manuscript now reads:

" . . . Note that sun glint observations were automatically filtered out because only the 10% observations having the lowest scene LER values were allowed to participate. . . . "

*l.242 – how does the proposed thresholding work over the solar-glint areas? Please provide more details.*

At this stage in the processing, sun glint observations should no longer be present because these were filtered out in the previous step described in section 4.6. If for some reasons a grid cell would suffer from sun glint at this stage, then this would be detected and remedied by the post-processing step. However, in practice, sun glint observations do not slip through to the post-processing stage.

We have added this to section 4.8 of the manuscript:

" Contamination by sun glint should not be present at this stage of the processing, because sun glint situations were filtered out quite robustly in the processing step described in Sect. 4.6. If for some reason contamination by sun glint would reach the post-processing step, then this would be detected and treated by the post-processing step in the same way as cloud contamination would be. "

*Fig. 5 – the adopted false-color scheme does not serve the purpose. With the exceptions of some cloud/aerosol contamination over the open-water areas, no effects (the claimed cloud-shadow contamination inclusive) are perceivable over the continental landmasses, since one may not be able to distinguish between the seasonal and the contamination-induced trends: e.g., Brazil/Amazonia in (a) vs. (c). Moreover, thus presented, the potential contamination cannot be quantified in any meaningful way. Instead, one may consider providing global maps showing estimates of the contamination magnitudes for some key (trace-gas retrievals, clouds) wavelengths, preferably expressed as percentages of the underlying reflectivities.*

It is true that cloud-shadow contamination cannot be seen in Figure 5. In earlier versions of the database, which did not have the cloud shadow filtering implemented, these were visible. The current version of the database does not suffer from these features. Unfortunately, we forgot to update the caption of Figure 5.

We have removed the wrong statement "In window (b) signs of cloud shadow impact can be seen in the southern regions" from the caption. To be precise, the caption of Figure 5 now reads:

" **Figure 5.** False colour composite images, created using the 402, 494, and 670-nm TROPOMI surface LER values, for four calendar months. While producing the images, the surface reflectivity was increased to better emphasise the presence of cloud and aerosol contamination, leading to saturation and discolouration over some of the desert areas. Certain regions show signs of cloud contamination. The impact of persistent aerosol plumes can also been seen in some cases. "

As for the cloud and aerosol contamination, these false-colour images were quite useful in the past when this method was applied to the GOME-1 LER database [Koelemeijer et al., 2003, Fig. 7], to the OMI LER database [Kleipool et al., 2008, Fig. 3], and to the SCIAMACHY and GOME-2 LER databases [Tilstra et al., 2017, Fig. 6]. The main reason for this is that these database were suffering more from cloud contamination, mainly because of the much larger footprint sizes of the observations, making the cloud contamination easier to observe in the false-colour images. A second reason is that for the retrieval of the GOME-1 and OMI LER databases no (absorbing) aerosol filtering was applied. Admittedly, in Figure 5 the cloud and aerosol contamination features are sometimes hard to spot, especially over the continental landmasses.

However, with the knowledge of the whereabouts of cloud and aerosol contamination hot spots obtained from the earlier LER databases, and by sharing this knowledge in the manuscript, we think it should be possible to observe the most prominent cloud and aerosol contamination features in Figure 5 also over the continental landmasses. The primary goal of Figure 5 is to show that cloud and aerosol contamination still exists to some degree, but that it is less of an issue compared to the earlier LER databases.

The strength of the false-colour method is that it does not require knowledge of the absolute magnitude of the cloud and aerosol contamination. Cloud contamination will always show up in white or grey. For the above reasons, and because the false-colour image in Figure 5 may be compared to similar false-colour images in the three papers mentioned above, we decided to keep Figure 5 the manuscript.

As for providing estimates of the cloud contamination, it is not possible to provide these because that would mean we would be able to know what the non-cloud contaminated situation would be.

*l.466, 469, etc. – please clarify the meaning of '0.03+10%'.*

The accuracy requirement is 0.03 plus an additional 10% of the value, so if the value is 0.1, then the uncertainty is 0.03 + 0.01 = 0.04 in the absolute sense.

Please note that, upon request by Reviewer #1, these uncertainty requirements are now mentioned in the introduction of the revised manuscript. The text in the introduction now reads:

" . . . For the validation study we used accuracy requirements on the DLER of 0.03+10% (0.03 plus 10% of the value, below 500 nm) and 0.02+10% (above 670 nm). . . . "

*l.464 and on – please comment, one more time, on the relatively large differences seen in the TROPOMI-GOME-2 comparisons (Table S4).*

Agreed, we have changed the text in section 6.3 in accordingly:

" . . . Comparing with the GOME-2 surface LER database, we find an offset of about −0.02 below 400 nm, which we can attribute to an offset in the GOME-2 surface LER database that was already reported earlier [Tilstra et al., 2017]. . . . "

*l.118 – ...course...*

Corrected.

*The Supplement:*

*Is there any reason behind the very different seasonal trends seen in the Asian (sub)-tropical forests (Figures S1-S3)? The difference between 463 nm and the adjacent 670 and 380 nm samples is rather striking. One may call the May 380 nm outlier a suspect (instrument – unmitigated straylight? clouds?). This difference is even more striking considering the close match between the average Asian-forest spectra plotted in Figure S4. The same may apply to the Evergreen woodland at 380 nm (August) as seen in Figures S1-S3.*

Thank you very much for reporting this. In fact, looking for an explanation for these issues we found a bug in our plotting routine. This bug was introduced at some point when the plotting routine was extended to also handle the 380, 494 and 670-nm wavelength bands meant for the supplement (next to the 772-nm wavelength band that was already presented in the manuscript and which was handled correctly). We have recreated the plots presented in Figures S1–S3 and this fortunately solves all the observed issues in one go.

*Figure S13 may help to estimate the magnitude of cloud contamination in the continental North America DLER data (April-September biases). This important estimate could be mentioned in the main text.*

We do see that there is a tendency towards higher slopes in the months April to September. Perhaps cloud contamination is an issue here. It could also be caused by aerosols. This is certainly something to look into. We have decided to mention the observation of the biases (higher slopes) in the Supplement and to note that

these may be related to cloud and/or aerosol contamination in the TROPOMI DLER database. The text in section S4.1 of the Supplement now reads:

" For most of the case studies there is not a large dependence on the calendar month. For case 3, the "North America" region, the scatter plots shown in Fig. S13 for the months April till September show slopes considerably larger than one. This may be related to cloud and/or aerosol contamination present in the TROPOMI DLER database and will be studied more closely in the future. "

**References:**

Kleipool, Q. L., Dobber, M. R., de Haan, J. F., and Levelt, P. F.: Earth surface reflectance climatology from 3 years of OMI data, J. Geophys. Res., 113, D18308, doi:10.1029/2008JD010290, 2008.

Kleipool, Q., Ludewig, A., Babić, L., Bartstra, R., Braak, R., Dierssen, W., Dewitte, P.-J., Kenter, P., Landzaat, R., Leloux, J., Loots, E., Meijering, P., van der Plas, E., Rozemeijer, N., Schepers, D., Schiavini, D., Smeets, J., Vacanti, G., Vonk, F., and Veefkind, P.: Pre-launch calibration results of the TROPOMI payload on-board the Sentinel-5 Precursor satellite, Atmos. Meas. Tech., 11, 6439–6479, doi:10.5194/amt-11-6439-2018, 2018.

Koelemeijer, R. B. A., de Haan, J. F., and Stammes, P.: A database of spectral surface reflectivity in the range 335–772 nm derived from 5.5 years of GOME observations, J. Geophys. Res., 108, 4070, doi:10.1029/2002JD002429, 2003.

Ludewig, A., Kleipool, Q., Bartstra, R., Landzaat, R., Leloux, J., Loots, E., Meijering, P., van der Plas, E., Rozemeijer, N., Vonk, F., and Veefkind, P.: In-flight calibration results of the TROPOMI payload on board the Sentinel-5 Precursor satellite, Atmos. Meas. Tech., 13, 3561–3580, doi:10.5194/amt-13-3561-2020, 2020.

Stein Zweers, D. C.: TROPOMI ATBD of the UV aerosol index, Doc. No. S5P-KNMI-L2-0008-RP, Issue 2.1.0, 22 July, Koninklijk Ned. Meteorol. Inst., De Bilt, the Netherlands, available at: https://sentinels.copernicus.eu/documents/247904/2476257/Sentinel-5P-TROPOMI-ATBD-UV-Aerosol-Index.pdf, 2022.

Tilstra, L. G., Tuinder, O. N. E., Wang, P., and Stammes, P.: Surface reflectivity climatologies from UV to NIR determined from Earth observations by GOME-2 and SCIAMACHY, J. Geophys. Res.-Atmos., 122, 4084–4111, doi:10.1002/2016JD025940, 2017.

Tilstra, L. G., de Graaf, M., Wang, P., and Stammes, P.: In-orbit Earth reflectance validation of TROPOMI on board the Sentinel-5 Precursor satellite, Atmos. Meas. Tech., 13, 4479–4497, doi:10.5194/amt-13-4479-2020, 2020.

Veefkind, J. P., Aben, I., McMullan, K., Förster, H., de Vries, J., Otter, G., Claas, J., Eskes, H. J., de Haan, J. F., Kleipool, Q., van Weele, M., Hasekamp, O., Hoogeveen, R., Landgraf, J., Snel, R., Tol, P., Ingmann, P., Voors, R., Kruizinga, B., Vink, R., Visser, H., and Levelt, P. F.: TROPOMI on the ESA Sentinel-5 Precursor: A GMES mission for global observations of the atmospheric composition for climate, air quality and ozone layer applications, Remote Sens. Environ., 120, 70–83, doi:10.1016/j.rse.2011.09.027, 2012.

---

## Author Comment (AC2)

**Response to RC1 by Reviewer 1:**

We would like to thank the reviewer for performing a thorough review and for the many helpful suggestions to improve our manuscript.

Below, we respond to each of the review comments. For the sake of clarity, the review comments are given in blue italics and our response is printed in normal font. Changes to the manuscript are printed in green.

*I found this paper to be carefully constructed and well written. The authors provide a clear motivation for the work, and did a good job explaining details that might not be obvious to all readers. For the most part the reader need not be familiar with previous work to understand and follow the discussion in this paper.*

*Section 1*

*The authors fail to discuss the version of the TropOMI Level 1B product used in this work until Section 6.3. The version should be cited early in the paper along with the doi of the data. As the authors note, the TOA reflectance changes between product versions so it's important to state key facts early.*

We agree and have added the version number of the TROPOMI L1 data to the introduction of the manuscript. The DOI of the L1B data (10.5270/S5P-kb39wni) is now also available as a link.

The text in section 1 of the manuscript now reads:

" For the generation of the DLER database we used TROPOMI level-1b data version 2.1.0 (doi:10.5270/S5P-kb39wni). "

*A similar criticism can be made regarding accuracy requirements. The requirements for this work best belong in the introduction where their origin can also be described.*

We now mention the accuracy requirements already in the introduction. Their origin is described in the Final Report of the S5p+ Innovation project which was set up by ESA to support the development of several TROPOMI products. We refer to this report in the revised version of the manuscript.

The text in section 1 of the manuscript now reads:

" For the validation study we used accuracy requirements on the DLER of 0.03+10% (0.03 plus 10% of the value, below 500 nm) and 0.02+10% (above 670 nm). These target requirements were taken from the final report of ESA's Sentinel-5p+ Innovation AOD/BRDF project [Litvinov et al., 2022]. "

*Use of a DLER product from TropOMI is pretty much limited to satellite observations in a 1330 orbit, possibly a few others if reciprocity is assumed. While there are a significant number of instruments orbiting at this time of day it still limits the application of these results. If the authors were to assume surface BRDF models (e.g. from MODIS) it should be possible to derive a total hemispheric reflectance (THR) from these measurements. A THR product can broaden the reach of these data, allowing a larger pool of potential comparisons including instruments in morning polar orbits and geostationary instruments. The authors have effectively already performed such a comparison between afternoon orbit TropOMI data and the morning orbit MODIS data. Just a thought for a future paper.*

The use of the TROPOMI DLER is indeed limited to satellite observations in a 13:30 orbit, or orbits close to that. Using a kernel-based approach such as was used for MODIS BRDF was considered by us in the past, but this is difficult to do properly with observations originating from only one overpass time. For the derivation of the MODIS BRDF product, data from morning (Terra) and afternoon (Aqua) orbits are used, such that the derivation of the kernel coefficients is supported by at least two domains of solar geometries. This is not possible for TROPOMI. Ideas to combine future Sentinel-5/UVNS data (equator crossing time of 09:30 LT) with TROPOMI data exist, but the launch of Sentinel-5 is not foreseen until 2025.

*Section 4.5*

*It would be useful to know how sensitive LER is to the AI screening level as a way of evaluating the chosen threshold. Have the authors performed such a study? Can they provide some more justification for the screening threshold of AI = 2?*

The choice for setting the threshold is partly subjective. In the paper introducing the OMI GLER paper [Qin et al., 2019], a threshold of 1.0 is used. The decision to use a more relaxed threshold of 2.0 index points was made by us because we think it is not essential to remove all levels of aerosol, especially not when this is achieved at the cost of a reduction in quality. In other words, filtering out too much will lead to gaps, large uncertainties, or biases which we want to avoid at all costs.

There are other reasons for being conservative. A trend in the AAI values could potentially harm the retrieval or bias it to certain years. The TROPOMI AAI product shows increased values at the edges of the swath. Also, the calibration of the TROPOMI AAI product has changed from version to version, so the AAI threshold to be used depends on the actual version of the TROPOMI AAI product used. It should be mentioned, though, that the latest version of the TROPOMI AAI product (v2.6.0) includes a degradation correction and is better calibrated than earlier versions.

In any case, we performed tests to see which threshold would work best, just by inspecting the fields that were returned. In the end we opted, more or less by coincidence, for the same threshold we use for the GOME-2 surface DLER database (2.0 index points). This particular threshold will remove the strongest cases of aerosol presence, but it will, admittedly, not remove small background levels of aerosol.

A plan for the future is to improve the filtering on aerosol by using AOD information. This filtering on AOD could be performed in addition to the more conservative filtering on AAI.

*Sections 4.6 & 4.7*

*In these two sections the authors are perhaps a bit too reliant on the reader having read and remembered their previous publication on this topic. The reader is left wondering about the general approach. For example, an elaborate method of selecting a representative LER for each grid is described in Section 4.6. No mention is given to view angles, so one must assume that all angles are included in the final distributions. However, a selection of values is then made based on the lowest 10% (or the mode of the distribution in the case of snow/ice). Such a selection is necessarily biased toward view angles where the BRDF is at a minimum, meaning the Section 4.6 LER is dependent on viewing conditions. In Eqn. 8 a quantity ALER is introduced for the first time with no explanation of where it comes from. The reader can deduce from Lines 237-239 that ALER is*

*the reflectivity of water, which is not a Lambertian quantity except in ideal conditions. How does the LER of Section 4.6 relate to Eqn. 8? Section 4.7 stands out in its need for clearer explanation when compared with the others parts of this paper.*

Thank you for pointing this out to us. Indeed, the text in section 4.6 does not mention the separation in viewing angle regimes that is introduced later in section 4.7. In principle section 4.6 only introduces the method used for the standard non-directional LER. In section 4.7 the DLER approach is introduced, which involves calculating the LER for individual viewing angle segments.

Indeed, the non-directional LER is biassed towards viewing angles for which the BRDF is at a minimum. This is a fundamental property of the non-directional LER as retrieved in the past from instruments like TOMS, GOME-1, OMI, and SCIAMACHY. This is explained in the introduction of the paper. The DLER does not suffer from this limitation, because it takes the angular dependence into account by cutting up the viewing angle range into segments and calculating the LER for each of these viewing angle segments. After this, the directional dependence can be determined.

To clarify the situation, we have decided to change section 4.6, by already mentioning the possibility to cut up the swath in viewing angle segments and making the link with section 4.7, where the DLER is introduced. We also make clear that section 4.6 introduces the traditional, non-directional LER database.

The text in section 4.6 of the manuscript now reads:

" The traditional, non-directional surface LER database is calculated in the following way. For each calendar month, the observations from all available mission years which are considered cloud-free, clouds shadow-free, and aerosol-free by the screening steps are mapped onto a 0.125 by 0.125 degrees latitude/longitude grid. In this step, all viewing angles are accepted, although the code can also be instructed to only take a certain viewing angle range into account. The latter possibility is not used here, but it will be used for the DLER calculation introduced in Sect. 4.7. The distribution of the scene LER values of each grid cell is then analysed . . . "

Additionally, we reworked section 4.7 to better explain the meaning of equation (8).

The text in section 4.7 of the manuscript now reads:

" . . . In the retrieval code, the viewing angle range available for this is cut up into nine viewing angle containers and the normal surface LER retrieval introduced in Sect. 4.6 is performed for each of these nine containers. This results in nine surface LER values, which, as a function of viewing angle $\theta_v$, are fitted by a third-order polynomial. The DLER can then be parameterised as a function of the viewing angle $\theta_v$, as was done in Tilstra et al. [2021], however, with a third order term:

$$A_{\mathrm{DLER}} = A_{\mathrm{LER}} + c_0 + c_1 \cdot \theta_v + c_2 \cdot \theta_v^2 + c_3 \cdot \theta_v^3. \tag{1}$$

In Eq. (8), the directional surface LER $A_{\mathrm{DLER}}$ is expressed as an extension on top of the non-directional surface LER $A_{\mathrm{LER}}$. The values of $A_{\mathrm{LER}}$ and of the polynomial coefficients $c_0$, $c_1$, $c_2$, and $c_3$ are contained in the database file. For water surfaces, the polynomial coefficients are set equal to zero. . . . "

*Section 5.4*

*The authors use a visual example to demonstrate the need for and the effectiveness of their cloud screening*

*method. In the example shown in Figure 6 it is not immediately obvious that every feature in black is a cloud shadow that should be removed. Shadows at this location should appear to the north north-east of the actual cloud, and it's rather difficult to imagine clouds that could produce some of the shadows seen to the north and north-east of Iceland in Figure 6a. It would be helpful if the authors could include a VIIRS RGB image or a TropOMI reflectivity image for the same time period to show the actual cloud field. There may be clever ways of including this as a transparency overlayed on the minimum reflectivity maps.*

Figure 6 presents retrieved surface LER, so clouds cannot be seen in this figure, because clouds have been filtered out by the two-step cloud filtering described in sections 4.3 and 4.6. As a result, it is impossible to know where the clouds responsible for the cloud shadows were located. It is, therefore, not possible to relate the position of the black features to the position of clouds. What we see are projections of the cloud shadows, not the clouds that produced them.

Moreover, the surface LER field presented in Figure 6 is determined from one month of data, so it is not possible to provide a map of the cloud field.

The cloud shadows in this location are generally located to the north of the clouds. However, because of the parallax effect, they can be found on the east and on the west side of the clouds as well. The parallax effect, caused by the east and west viewing geometries of the TROPOMI instrument, is demonstrated in Figure 9 of Trees et al. [2022]. Apart from the parallax effect, the shape of the cloud also determines the shape of the cloud shadow projected onto the surface.

Figure 6 does, therefore, not tell if the clouds shadows are located at the north, east, or west of the clouds. This cannot be deduced from the shape of the black features.

We have changed the text to clarify that the black features in Figure 6 are caused by cloud shadows. The text in section 5.4 of the manuscript now reads:

" … In the left window, individual cloud shadows can be seen as black features. These cloud shadows are taken into account by the retrieval algorithm which is focused on the lowest scene LER values … "

*Section 6.1*

*The authors should cite the product versions used in all the comparisons described in this section.*

We agree and now mention the product versions of the TROPOMI (v2.1), GOME-1 (v1.0), OMI (v3), SCIA-MACHY (v2.6), GOME-2 (v4.0), and MODIS (v6.1) surface reflectivity databases in the revised manuscript.

*Section 6.2.1, Lines 408-411*

*The authors cite Rayleigh scattering effects as a reason for not comparing DLER in the UV to the MODIS BRDF. To first order Rayleigh scattering should already be taken into account via the derivation of surface reflectivity given by Eqn. 2. The more important reason for not comparing in the UV is that MODIS makes no measurements at these wavelengths and few, if any, estimates of BRDF exist at these wavelengths. It's worth noting that a UV comparison in conjunction with a long-wave VIS comparison to the MODIS BRDF could provide a useful assessment of how BRDF might change between VIS and the UV.*

We mentioned the Rayleigh optical thickness merely to point out that there is considerable multiple scattering

for the shorter wavelengths. The increase in multiple scattering is the reason why DLER and BRDF are different below ∼500 nm, as explained in a previous paper by us [Tilstra et al., 2021]. For this reason, we do not attempt to compare the DLER with MODIS BRDF for its shortest (469-nm) wavelength band.

We'd like to point out, though, that we have, in the mean time, added a comparison with the OMI GLER product for this particular wavelength band. This comparison was suggested by Reviewer #2. The results of this comparison are now discussed in the new section 6.3 of the revised manuscript.

*Section 6.3, Lines 481, 482*

*The authors state that TropOMI calibration is much improved in the v2.1 Level 1 product compared to v1.0. Can they provide a reference to substantiate this claim? There exists evidence that the TOA reflectance in the latter version is actually less accurate than in the earlier version. Figure 10d in particular is suggestive of a calibration difference of 2%.*

The absolute radiometric calibration was studied by several parties, also by us. Results for the previous version of the L1B data were reported in, for instance, Tilstra et al. [2020]. One of the conclusions was that there is a ∼10% radiometric calibration error for TROPOMI bands 3/4 in version 1.0.0 of the L1B data. We have since then concluded that this calibration error went down to ∼0–2% in the latest version (v2.1.0) of the L1B data.

We were not aware that there exists evidence that the reflectance actually decreased in quality with the release of the latest version. It would be worthwhile to share this information with the TROPOMI calibration team.

**References:**

Litvinov, P., Chen, C., Dubovik, O., Fuertes, D., Bindreiter, L., Lanzinger, V., de Graaf, M., Tilstra, G., Stammes, P.: S5p+ Innovation AOD/BRDF Final Report, GRASP/KNMI, Issue 1.0, 10 February, available at: https://d37onar3vnbj2y.cloudfront.net/static/surface/albedo/documents/S5p%2BInnovation_AOD_BRDF_Final_Report_v1.1.pdf, 2022.

Qin, W., Fasnacht, Z., Haffner, D., Vasilkov, A., Joiner, J., Krotkov, N., Fisher, B., and Spurr, R.: A geometry-dependent surface Lambertian-equivalent reflectivity product for UV–Vis retrievals – Part 1: Evaluation over land surfaces using measurements from OMI at 466 nm, Atmos. Meas. Tech., 12, 3997–4017, doi:10.5194/amt-12-3997-2019, 2019.

Tilstra, L. G., de Graaf, M., Wang, P., and Stammes, P.: In-orbit Earth reflectance validation of TROPOMI on board the Sentinel-5 Precursor satellite, Atmos. Meas. Tech., 13, 4479–4497, doi:10.5194/amt-13-4479-2020, 2020.

Tilstra, L. G., Tuinder, O. N. E., Wang, P., and Stammes, P.: Directionally dependent Lambertian-equivalent reflectivity (DLER) of the Earth's surface measured by the GOME-2 satellite instruments, Atmos. Meas. Tech., 14, 4219–4238, doi:10.5194/amt-14-4219-2021, 2021.

Trees, V. J. H., Wang, P., Stammes, P., Tilstra, L. G., Donovan, D. P., and Siebesma, A. P.: DARCLOS: a cloud shadow detection algorithm for TROPOMI, Atmos. Meas. Tech., 15, 3121–3140, doi:10.5194/amt-15-3121-2022, 2022.

---

## Author Response (AR2)

**Author response**

We have resolved the following two issues in the final version of the manuscript:

*- Eq. 1: The viewing and azimuth angles are defined in the text, but are not included in Eq. 1., where I expect that I and R depend on both angles.*

The dependence on mu, mu0, phi and phi_0 is now mentioned in Eq. 1, in both the I and the R.

*- Line 148: explain acronym ATBD.*

We have added the meaning of the acronym ATBD:

"... can be found in the Algorithm Theoretical Baseline Document (ATBD) ..."